# Genome-Wide Identification and Analysis of the Genes Encoding Q-Type C2H2 Zinc Finger Proteins in Grapevine

**DOI:** 10.3390/ijms242015180

**Published:** 2023-10-14

**Authors:** Mingyu Chu, Tiaoye Wang, Wenfang Li, Yashi Liu, Zhiyuan Bian, Juan Mao, Baihong Chen

**Affiliations:** College of Horticulture, Gansu Agricultural University, Lanzhou 730070, China; chu.my@foxmail.com (M.C.);

**Keywords:** grapevine, abiotic stress, Q-type C2H2 zinc finger proteins, qRT-PCR

## Abstract

Q-type C2H2 zinc finger proteins (ZFPs), the largest family of transcription factors, have been extensively studied in plant genomes. However, the genes encoding this transcription factor family have not been explored in grapevine genomes. Therefore, in this study, we conducted a genome-wide identification of *ZFP* genes in three species of grapevine, namely *Vitis vinifera*, *Vitis riparia*, and *Vitis amurensis*, based on the sequence databases and phylogenetic and their conserved domains. We identified 52, 54, and 55 members of Q-type C2H2 *ZFPs* in *V. vinifera*, *V. riparia*, and *V. amurensis*, respectively. The physical and chemical properties of VvZFPs, VrZFPs, and VaZFPs were examined. The results showed that these proteins exhibited differences in the physical and chemical properties and that they all were hydrophobic proteins; the instability index showed that the four proteins were stable. The subcellular location of the *ZFPs* in the grapevine was predicted mainly in the nucleus. The phylogenetic tree analysis of the amino acid sequences of VvZFP, *VaZFP*, *VrZFP*, and *AtZFP* proteins showed that they were closely related and were divided into six subgroups. Chromosome mapping analysis showed that *VvZFPs*, *VrZFPs*, and *VaZFPs* were unevenly distributed on different chromosomes. The clustered gene analysis showed that the motif distribution was similar and the sequence of genes was highly conserved. Exon and intron structure analysis showed that 118 genes of *ZFPs* were intron deletion types, and the remaining genes had variable numbers of introns, ranging from 2 to 15. Cis-element analysis showed that the promoter of *VvZFPs* contained multiple cis-elements related to plant hormone response, stress resistance, and growth, among which the stress resistance elements were the predominant elements. Finally, the expression of *VvZFP* genes was determined using real-time quantitative PCR, which confirmed that the identified genes were involved in response to methyl jasmonate (MeJA), abscisic acid (ABA), salicylic acid (SA), and low-temperature (4 °C) stress. *VvZFP10*-GFP and *VvZFP46*-GFP fusion proteins were localized in the nucleus of tobacco cells, and *VvZFP10* is the most responsive gene among all *VvZFPs* with the highest relative expression level to MeJA, ABA, SA and low-temperature (4 °C) stress. The present study provides a theoretical basis for exploring the mechanism of response to exogenous hormones and low-temperature tolerance in grapes and its molecular breeding in the future.

## 1. Introduction

Zinc finger proteins (ZFPs) constitute a large family of transcription factors that are distributed widely in plants. A common feature of ZFPs is that they can stabilize a very short multi-foldable self-folding finger peptide space configuration through their coordinated Zn ions (Zn^2+^). Presently, the most widely distributed and well-studied ZFP family genes in eukaryotic genomes are those encoding the C2H2-ZFPs, which are also known as the TFIIIA-type ZFPs. These genes account for nearly 0.7% of the total number of plant genes. C2H2 zinc finger proteins that function as transcript factors (TFs)typically contain an array of two or more tandemly arranged C2H2 motifs; each C2H2 zinc finger protein in such polydactyl-fingered or “polydactyl” proteins can bind to three adjacent nucleotides at target sites, with amino acids at positions 1, 2, 3, and 6 in the alpha helical region of each motif having a crucial role in DNA recognition [1,2]. In addition, C2H2-ZFPs, as transcription factors, generally have a nuclear localization signal region, which is also known as B -box [3].“Q-type” C2H2 ZFPs, with the sequence QALGGH, can combine with the residues in the protein structure; specifically, they can combine with the promoter region of regulation-related genes and thus control the expression of genes in plants and play an important role in the growth, development, and stress response of plants [4,5].

Studies have found that *ZFPs* are crucial in the growth, development, and response to abiotic stress in cassava [6,7,8]. During flower development of Chinese cabbage, ZFPs participate in the flowering induction process as histone methyltransferase or demethylase and perform transcriptional regulation of characteristic genes of flower organ models by affecting cell proliferation and hormonal regulation [9]. Based on the expression patterns of *CsC2H2-ZFPs*, a study reported that these *ZFPs* were responsive to different stresses including drought, salt, cold, and methyl jasmonate (MeJA) treatments [10]. *OsZFP179* from rice was functionally characterized as a salt-responsive gene, and its overexpression was shown to enhance salt tolerance in transgenic rice plants [11].

The *ZFP* family has been implicated in the adaptation or response of plants to abiotic stresses. Some expression analyses have shown that various *ZFP* genes in Arabidopsis are induced in response to cold and drought stress [12]. Some *ZFP* genes such as *AtZAT12* are also involved in oxidative and abiotic stress signaling [12,13,14], and oxidative stress occurs under various biotic and abiotic stress conditions [15]. A review highlighted that *AZF2*, *STZ*, and *ZAT12*, which are C2H2 zinc finger genes from Arabidopsis, form a regulatory network of cold-responsive genes [16]. A cold-inducible gene, *SCOF-1*, from soybean, was reported to enhance cold tolerance [17]. Recent transgenic studies have shown that the overexpression of some *ZFP* genes from Arabidopsis and rice can enhance salt tolerance [15,18,19]. Likewise, overexpression of drought-inducible *ZFP* genes (*ZPT2-3* in Petunia and STZ in Arabidopsis) has been reported to enhance drought tolerance in transgenic plants [20,21]. Most *ZFP* genes involved in plant response to abiotic stress are transcriptional repressors [18,21,22] and the EAR motif of ZAT7 is critical for the ZFP-mediated improvement of abiotic stress tolerance in transgenic plants [18].

*ZFPs* have been extensively studied in plants; studies have identified 179 *ZFP* genes in Arabidopsis [23], 79 in potato [9], 47 in *Triticum aestivum* [6], 118 in tobacco [24], 120 in cucumber [25], 189 in rice [26], and 109 in poplar [27]; however, this transcription factor family genes in grapevine have not been explored. Therefore, the present study was conducted to identify the genes and explore the structure of the *ZFP* family in the genome of three species of grapevine, namely *Vitis vinifera*, *Vitis riparia*, and *Vitis amurensis*.

In this study, a genome-wide identification of *ZFP* genes in the three species of grapevine was performed based on the current sequence databases and subsequent phylogenetic and conserved domains. Subsequently, we explored the evolution of the *ZFP* genes by analyzing intragroup gene collinearity of the three grapevine genomes and interspecies gene collinearity among *V. vinifera* and other five plants, namely *Malus domestic*, *Prunus persica*, *Musa acuminate*, *Solanum lycopersicum*, and *Arabidopsis*. Furthermore, the mRNA distribution of these *VvZFPs* in various organs (roots, leaves, flowers, and seeds.) was determined. Our expression analyses showed that the *VvZFP* subfamily in grapevine is predominantly expressed in these organs. Further, to predict their potential functional mechanisms, quantitative analysis of the expression of *VvZFP* family genes in response to MeJA, abscisic acid (ABA), salicylic acid (SA), and low-temperature (4 °C) treatments was performed. The present study provides a theoretical basis for further research on the biological functions of *ZFP* genes in grapevine.

## 2. Results

### 2.1. Identification of Q-Type ZFP Gene Family in Grape Genome

The hidden Markov Model was used to identify the *ZFP* genes in grape genomes. A total of 52 Q-type *VvZFPs*, 54 Q-type *VrZFPs*, and 55 Q-type *VaZFPs* were identified, which were named *VvZFP1*–*VvZFP52*, *VrZFP1*–*VrZFP54*, and *VaZFP1*–*VaZFP55*, respectively, according to their chromosomal positions. The physical and chemical properties of the proteins encoded by *VvZFP*, *VaZFP*, and *VrZFP* genes are summarized in Table 1. Regarding *VvZFP* genes, the coding sequence (CDS) length was 468–1830 bp, the protein length was 155–609 amino acids, the molecular weight was 17,251.04–65,943.19 Da, pI was 4.79–9.58, and the instability index ranged from 38.51 to 77.98. All 52 VvZFP proteins were predicted to be localized in the nucleus, and their secondary structure comprised mainly the alpha helix, extended strand, beta-turn, and random coil. For *VaZFP* genes, the CDS length was 474–2583 bp, the protein length was 157–860 amino acids, the molecular weight was 17,102.01–94,567.56 Da, pI was 4.94–10, and the instability index ranged from 38.21 to 79.33. All VaZFP proteins except VaZFP24 were predicted to be localized in the nucleus; VaZFP24 was predicted to be localized in the cell membrane. The protein secondary structure comprised mainly the alpha helix, extended strand, and random coil. Regarding *VrZFP* genes (Table 1), the CDS ranged from 396 to 1836 bp, the protein length was 131–611 amino acids, the molecular weight was 14,365.07–66,157.24 Da, pI was between 4.78 and 9.3, and the instability index ranged from 37.24 to 80.72. All 53 VrZFP proteins except VrZFP10 were predicted to be localized in the nucleus; VrZFP10 was predicted to be localized in the cell membrane. Their protein secondary structure comprised mainly the alpha helix, extended strand, and random coil.

### 2.2. Phylogenetic Analysis of ZFP Proteins in Grape Genomes

The identified 52 Q-type VvZFP, 54 Q-type VrZFP, 55 Q-type VaZFP, and 57 Q-type AtZFP proteins were obtained from the Phytozome, VITSGDB, NCBI, and TAIR databases, respectively. Multiple sequence alignment of the 218 ZFP proteins was performed using the software MEGA7.0 [28]. According to the distance of their kinship, the 218 ZFP proteins were divided into eight groups (Figure 1). Group 1 comprised 5 *VvZFP*, 5 *VrZFP*, 5 *VaZFP*, and 11 AtZFP proteins. Group 2 comprised 6 *VvZFP*, 4 VrZFP, 10 VaZFP, and 4 AtZFP proteins. Group 3 comprised 7 VvZFP, 8 VaZFP, 8 VrZFP, and 9 AtZFP genes. Group 4 comprised 3 VvZFP, 4 VaZFP, 3 VrZFP, and 7 AtZFP proteins. Group 5 comprised 10 VvZFP, 9 VaZFP, 10 VrZFP, and 8 AtZFP proteins. Group 6 comprised 10 VvZFP, 9 VaZFP, 10 VrZFP, and 8 AtZFP genes. Group 7 comprised 6 *VvZFP*, 6 *VaZFP*, 8 *VrZFP*, and 6 AtZFP proteins. Group 8 comprised 12 VvZFP, 11 VaZFP, 11 VrZFP, and 9 AtZFP proteins.

### 2.3. Gene Structure and Conserved Motif Analysis of the ZFP

Using MEME, a total of 15 conserved motifs of the ZFP amino acid sequences were analyzed. The phylogenetic tree members on the same branch exhibited slight differences and those on different branches exhibited great differences in terms of the type, quantity, and position of the motifs (Figure 2A,B). The whole genome was divided into eight subgroups based on the phylogenetic relationship. Among the 15 motifs, the sequence of motif 1 was “YECNFCNREFPSSQALGGHQNAHKKER ARAKRSQ”, which is Zinc finger #1; the sequence of motif 2 was “KTHECSICSKEFSSGQALGGHMRC HRERE”, which is Zinc finger #2; the sequence of motif 3 was “EDEEEEEDLANCLIMLSRGGG [29]”, which is L-box motif; the sequence of motif 5 was “CSTCKKVFPSGQALGGHRRSH”, which is finger #3; the sequence of motif 6 was “DLDLNLRL”, which is an EAR motif; and the sequence of motif 13 was “EVWTKRKRSKRIRLD”, which is B-box. All members of Group 1 comprised mainly motifs 1, 2, 3, and 6. Among them, some members of Group 1 (VvZFP41, VvZFP23, VvZFP35, VaZFP39, VaZFP13, VaZFP28, VrZFP40, VrZFP19, and VrZFP35) contained an additional motif, that is, motif 13. Group 2 contained the largest number and variety of motifs, namely motifs 1, 2, 4, 5, 6, 8, 9, 12, and 14 (VvZFP21, VvZFP20, VvZFP18, VaZFP16, VaZFP18, VaZFP19, VaZFP14, VrZFP21 contained two additional motifs, that is, motifs 11 and 15; VvZFP19, VvZFP20, VvZFP22, VaZFP18, VaZFP17, VaZFP20, VrZFP20, and VrZFP25 comprised an additional motif, that is, motif 10; VvZFP51, VaZFP50, and VrZFP54 contained motifs 1, 2, 4, 5, 8, 12, and 14, as well as motifs 3 and 13. VaZFP25 contained motifs 1, 6, and 7; and VaZFP35 contained only motifs 1, 3, and 6). Group 3 comprised 6 motifs, namely motifs 1, 2, 3, 5, 6, and 13 (VvZFP7, VvZFP8, and VvZFP11 contained motifs 2, 3, 6, 12, and 13; VvZFP48 comprised motifs 2, 3, 6, and 13; and VvZFP52, VaZFP4, and VrZFP10 comprised motifs 2, 3, 5, 6, and 13, as well as motifs 14 and 15). Group 4 comprised 5 motifs, namely motifs 1, 2, 3, 6, and 13 (VaZFP12 contained motifs 1, 6, and 8; and VaZFP48 contained motifs 1, 6, 7, and 10). Group 5 comprised two motifs, namely motifs 1 and 6 (VvZFP13 and VrZFP17 contained an additional motif, that is, motif 8; VvZFP6, VvZFP40, VaZFP5, VaZFP38, VrZFP9, and VvZFP41 contained motifs 1, 6, and 15; and VvZFP17, VvZFP34, VaZFP21, VaZFP29, VaZFP52, VrZFP22, and VrZFP34 contained not only two additional that is, motifs 13 and 15). Group 6 (VvZFP2, VaZFP2, VrZFP2, and VrZFP3) comprised only one motif, that is, motif 1. Group 7 comprised mainly motifs 1 and 6 (VvZFP44, VaZFP42, VaZFP53, VrZFP44, and VrZFP50 comprised only motif 1; VvZFP5, VrZFP7, and VrZFP8 comprised an additional motif 13). 

Group 8 comprised mainly motifs 1, 6, and 7 (VvZFP29, VvZFP42, VaZFP26, VaZFP36, VrZFP26, VrZFP39 contained not only motifs 1, 6, and 7 but also motif 4; VvZFP37, VaZFP34, and VrZFP37 contained motifs 10 and 15, in addition to motifs 1, 6, and 7; VvZFP31, VaZFP32, and VrZFP29 contained an additional motif 14; VaZFP41 contained only motif 1; and VvZFP38 and VrZFP38 contained only motifs 1 and 6).

To further understand the diversity of gene structure, the composition of introns and exons of genes was analyzed. As shown in Figure 2C, among the 161 members of *VaZFP*, *VvZFP*, and *VrZFP*, 118 genes were intron deletion types, only contained one exon, accounting for 74.5% of all members, 131 genes lacked upstream and downstream structures, 24 genes contained two exons. The number of exons in the remaining members (19 genes) ranged from 3 to 15, and the genes with the highest number of exons were *VaZFP35* and *VaZFP24*, with 15 exons. *VaZFP* 48 contains 12 exons, *VaZFP6* contains eight exons, *VaZFP24* and *VvZFP48* genes contain seven exons, *VvZFP2* and *VaZFP2* contain six exons, *VrZFP2* and *VrZFP3* contain five exons. Relatively speaking, genes with a higher number of exons are dominated by *VaZFP*. This may be related to the differentiation of gene function. Because the number and distribution of exons may affect the function and expression regulation of genes.

### 2.4. Analysis of Cis-Acting Elements of VvZFP, VrZFP, and VaZFP Genes

Analysis of the upstream 2000-bp cis-acting elements of the *VvZFP*, *VrZFP*, and *VaZFP* genes showed that nine types of hormone- and stress-related cis-acting regulatory elements were present in the promoters of ZFP genes in grape genomes (Figure 3 and Appendix A). Stress-related cis-acting elements (n = 3) included TC-rich repeats (defense and stress), MBS (drought), and low temperature-response (LTR) elements, whereas hormone-related cis-acting elements (n = 6) included TGA element/AuxRR core (auxin), O_2_ site (zein metabolism), TCA element/SARE (salicylic acid), ABRE (abscisic acid-response element), GARE-motif/P-box/TATC-box (gibberellic acid-response element), and CGTCA/TGACG motif (MeJA-response element). In addition, seven growth and development-related elements were identified including ARE (anaerobic induction), circadian control, RY-element (seed-specific regulation), CAT-box (meristem-specific expression), WUN-motif (wound-responsive element), MBSI (MYB binding site involved in the regulation of flavonoid biosynthesis genes), and AT-rich sequence.

Overall, 94% of the *ZFP* family members were found to contain TATA-box and CAAT-box core elements; 84.6% of the family members contained hormone response-related elements. Of these 84.6%, 55.7% were found to contain ABA-responsive elements (ABRE), 15.3% contained the AuxRR-core (auxin-responsive element), 7.7% contained the GARE-motif (gibberellin-responsive element), and 44.2% contained the TCA-element and salicylic acid-response element. Furthermore, 69.2% of the family members contained stress-response elements; of these, 17.3% contained the LTR element, 44.2% contained the MBS drought-response element, and 30.8% contained the TC-rich, defense- and stress-response elements.

### 2.5. Chromosomal Localization of ZFP Genes in the Grape Genome

To investigative the genomic distribution of *ZFP* genes on the chromosomes, the physical positions of all *ZFP* genes from *V. vinifera*, *V. riparia*, and *V. amurensis* were determined using the position data of ZFP genes obtained from the Ensembl, VITSGDB [30], and NCBI databases.

In *V. vinifera*, 52 *ZFP* genes were unevenly distributed on 18 chromosomes; *VvZFP* genes were located on Chr1–Chr9, Chr11, Chr13–Chr19, and ChrUn, with the least number (1) of genes located on Chr2, Chr4, Chr9, Chr11, Chr16, Chr17, Chr19, and ChrUn, and the highest number of genes (9) located on Chr6 (Figure 4A). In *V. amurensis*, 55 *ZFP* genes were unevenly distributed on 17 chromosomes, that is, Chr1–Chr9, Chr11, and Chr13-Chr19. *VaZFP51* and *VaZFP52* were located in scafold_187; *VaZFP53*, *VaZFP54*, and *VaZFP55* were located in scafold_503, scafold_1201, and scafold_1769, respectively. Chr4, Chr9, Chr11, Chr16, and Chr19 contained the minimum number of *VaZFP* genes (1), whereas Chr6 contained the highest number of *VaZFP* genes (10) (Figure 4B). In *V. riparia*, *54 ZFP* genes were unevenly distributed on 17 chromosomes, that is, Chr1-Chr9, Chr11, and Chr13- Chr19 chromosomes. The minimum number of *VrZFP* genes (1) were located on Chr4, Chr9, Chr11, and Chr16, whereas the highest number of *VrZFP* genes (8) were located on Chr13 (Figure 4C).

### 2.6. Analysis of Duplication, Ka/Ks, and Codon Usage Bias of ZFP Genes in Grape Genome

To further investigate the expansion pattern of *ZFP* genes in *V. vinifera*, *V. riparia*, and *V. amurensis*, the synteny analysis was carried out (Figure 5A–C). Intragenomic collinearity of *V. vinifera* was employed dada of the Plant Ensemble database. The intragroup collinearity analysis of *VvZFP* genes revealed 19 pairs of collinear genes (Figure 5A). These genes were located on chromosomes Chr1, Chr3, Chr4, Chr5, Chr6, Chr7, Chr8, Chr 9, Chr13, Chr14 and Chr18. Particularly, three genes on chromosome Chr6 were collineated with three genes on chromosome Chr8, respectively. Additionally, there are two genes on Chromosomes Chr3 that exhibited collinearity with the genes on chromosomes Chr7 and Chr18, respectively, namely *VvZFP8*/*VvZFP28*/*VvZFP50* and *VvZFP9*/*VvZFP27*/*VvZFP49*. Interestingly, three pairs of genes (*VvZFP15*/*VvZFP39*, *VvZFP17*/*VvZFP34* and *VvZFP23*/*VvZFP35*) showed a similarity of over 50% between their amino acid sequence. Furthermore, these two pairs of genes exhibited gene structure and conserved motifs.

The intragroup collinearity analysis of *VrZFP* genes revealed 18 pairs of collinear genes (Figure 5B), which were located on chromosomes Chr1, Chr2, Chr3, Chr6, Chr7, Chr8, Chr13, Chr14, Chr15, Chr16, Chr17, and Chr18. In addition, the two genes adjacent to each other on chromosome Chr3 were found to be collinear with the two genes adjacent to each other on chromosomes Chr7 and Chr18, respectively. These gene pairs were *VrZFP12*/*VrZFP27*/*VrZFP53* and *VrZFP13*/*VrZFP28*/*VrZFP52*. Four genes on chromosome Chr6 have observed the collinearity relationship with three genes on chromosome Chr8 and one gene on chromosome Chr13. Furthermore, the homogeneous analysis demonstrated that 16 pairs of collinear genes had high sequence homology (over 50%), and six pairs exceeding 60%. The intragroup collinearity analysis of *VaZFP* genes revealed nine pairs of collinear genes (Figure 5C), which were located on chromosomes Chr5, Chr6, Chr8, Chr11, Chr15 and Chr18, and four genes linked with other genes did not belong to the *ZFP* family. Among the collinear genes in *V. amurensis*, only one pair of genes had a sequence similarity higher than 50% (60.59%). These results indicated that segment duplication events have occurred during evolution in *VvZFP*, *VrZFP* and *VaZFP*. The expansion of the ZFP gene family may have occurred through duplication events, with segmental duplication playing a crucial role in the abundance of this gene family in the grapevine. Moreover, some segmental duplicated genes with high sequence homology shared a similar gene structure and conserved motifs. It is speculated that these genes may have a common ancestral gene and similar biological functions.

Codon bias analysis is useful in studying the evolution and environmental adaptability of species. The effective number of codons (Nc) and CAI of the *VvZFP*, *VaZFP*, and *VrZFP* genes were analyzed (Appendix A). In *V. vinifera*, the Nc value ranged from 44.69 (*VvZFP1*) to 61 (*VvZFP3* and *VvZFP25*), whereas the CAI value ranged from 0.146 (*VvZFP8*) to 0.276 (*VvZFP49*). In *V. amurensis*, the Nc value ranged from 45.84 (*VaZFP46*) to 61 (*VaZFP10*), and the CAI value ranged from 0.147 (*VaZFP2*) to 0.28 (*VaZFP49*). In *V. riparia*, the Nc value ranged from 44.6 (*VrZFP1*) to 61 (*VrZFP4*), and the CAI value ranged from 0.144 (*VrZFP12*) to 0.282 (*VrZFP52*). The relative synonymous codon usage (RSCU) values were employed to visually assess condon usage bias. Here, we analyzed the RSCU values of the VvZFP, VrZFP, and VaZFP proteins, as Appendix A show that the number of ZFP proteins in *V. amurensis* preferentially using codons with RSCU values >1 were higher than that of ZFP proteins in *V. vinifera* and ZFP proteins in *V. riparia*. In addition, the results of the correlation between codon usage parameters revealed that T3s were positively correlated with C3s, G3s, GC3s, CBI, and FOPs, whereas C3s were negatively correlated with CBI, FOPs, GC, and GC3s in *V. vinifera*, *V. riparia*, and *V. amurensis* (Figure 5D–F). Moreover, there is no difference in codon usage parameters among these three grapes.

The evolutionary selection pressure can be estimated based on the ratio Ka/Ks. Therefore, we calculated this value to further understand the evolutionary relationship among 

*VvZFP*, *VaZFP*, and *VrZFP* genes (Figure 5G–I, Appendix A). In *V. vinifera*, the Ka/Ks values of 639 gene pairs were calculated; 293 pairs exhibited the Ka/Ks of >1, two gene pairs (*VvZFP33*/*VvZFP1* and *VvZFP49*/*VvZFP4*) exhibited the Ka/Ks = 1, and 344 gene pairs exhibited the Ka/Ks of <1 (Figure 5G, Appendix A). In *V. riparia*, the Ka/Ks values of 779 gene pairs were calculated; 350 gene pairs exhibited the Ka/Ks > 1, seven gene pairs (*VaZFP44*/*VaZFP36*, *VaZFP37*/*VaZFP7*, *VaZFP36*/*VaZFP8*, *VaZFP32*/*VaZFP40*, *VaZFP33*/*VaZFP11*, *VaZFP31*/*VaZFP40*, and *VaZFP7*/*VaZFP10*) was equal to 1, and that of 234 gene pairs were less than 1. These results showed that during evolution, the *ZFP* gene families might have been subjected to purification selection in *V. vinifera* but to positive selection in *V. amurensis* and *V. riparia*.

To further understand the evolutionary relationship and gene function among the different species, we analyzed the interspecies collinearity between the *VvZFPs* and their counterparts in other five species, namely *Arabidopsis thaliana*, *Malus domestica*, *Prunus persica*, *Musa acuminata*, *Solanum lycopersicum* (Figure 6 and Appendix A), which revealed 326 pairs of orthologous genegenes, including 56 pairs of Arabidopsis genes, 64 pairs of peach genes, 106 pairs of apple genes, 40 pairs of banana genes, and 60 pairs of tomato genes. The homology relationship of *VvZFPs* with the apple genes was the strongest, followed by peach genes, whereas the relationship between *VvZFPs* and banana genes was the weakest, probably reflecting the evolutionary relationship and genetic specificity among species.

### 2.7. Analysis of VvZFP Gene Expression

The expression levels of *VvZFPs* in different tissues indicated similarities between their expression pattern in the same subgroup (Figure 7 and Appendix A). In Group 1, *VvZFP15* was downregulated in the burst initial stage of buds and seeds of post fruit set, but upregulated in other tissues; *VvZFP41* was upregulated in the flowers, pollens, and stamens. In Group 2, *VvZFP51* was upregulated in the seeds of post fruit set. In Group 3, *VvZFP45* was upregulated in the flowers, pollens, and stamens. In Group 4, *VvZFP49* was downregulated in buds at the burst initial stage and seeds at post fruit set stage but upregulated in other tissues. In Group 5, *VvZFP14* was upregulated only in the tendrils. In Group 6, *VvZFP2* was upregulated in young flowers and bud burst initial stage but downregulated in other tissues. In Group 7, *VvZFP5* was upregulated in the roots, stems, skin, flesh, rachis, pericarp, and seeds. In Group 8, *VvZFP4* was upregulated in the roots, stems, skin, flesh, flowers, petals, pollens, rachis, tendrils, carpels, buds, leaves, and seeds. Moreover, in this group, *VvZFP38* was upregulated in buds at bud burst initial stage, bud burst later stage, young leaves, and seeds but downregulated in other tissues.

### 2.8. qRT-PCR Analysis of the VvZFP

To elucidate the response of the *VvZFP* family in grapevine under extra hormone and stress, quantitative real-time PCR (qRT-PCR) was performed to characterize the expression pattern of all *VvZFPs* under MeJA, ABA, SA and 4 °C.

#### 2.8.1. Expression of *VvZFP* Genes under MeJA Treatment

After treatment with 100 μmol·L^−1^ MeJA, the genes that showed significant upregulation were *VvZFP5*, *VvZFP12*, *VvZFP13*, *VvZFP14*, *VvZFP16*, *VvZFP17*, *VvZFP19*, *VvZFP 22*, *VvZFP23*, *VvZFP25*, *VvZFP29*, *VvZFP30*, *VvZFP32*, *VvZFP34*, and *VvZFP43* (*p* < 0.05). At 3 h after MeJA treatment, the significantly upregulated genes were *VvZFP1*-*VvZFP6*, *VvZFP8*, *VvZFP9*, *VvZFP11*-*VvZFP23*, *VvZFP25*-*VvZFP30*, *VvZFP30*, *VvZFP35*, *VvZFP37*, *VvZFP39*, *VvZFP44*-*VvZFP46*, *VvZFP48*, *VvZFP49*, and *VvZFP52*. Among them, *VvZFP52* exhibited the highest relative expression level of 58.94 (Figure 8), 15 genes reached their expression peaks at 3 h, and the expression of *VvZFP3*, *VvZFP8*, *VvZFP9* and *VvZFP48* then dramatically decreased after 3 h, while others genes gradually decreased over time. Under MeJA treatment, the genes whose expression was significantly increased at 6 h were *VvZFP1*-*VvZFP5*, *VvZFP10*-*VvZFP31*, *VvZFP35*, *VvZFP37*, *VvZFP39*, *VvZFP40*, *VvZFP43*-*VvZFP46*, *VvZFP49*, and *VvZFP50*-*VvZFP52* (*p* < 0.01), among these 38 genes, *VvZFP10* exhibited the highest relative expression level of 6824.21. In addition, 21 genes have shown the expression peak at 6 h. For 12 h, the genes whose expression was increased significantly included *VvZFP3*, *VvZFP5*, *VvZFP6*, *VvZFP9*, *VvZFP11*-*VvZFP17*, *VvZFP22*-*VvZFP25*, *VvZFP29*-*VvZFP32*, *VvZFP35*, *VvZFP38*, and *VvZFP41*-*VvZFP43*, of which *VvZFP38* had the highest relative expression level of 372.04; furthermore, the expression levels of six genes (*VvZFP6*, *VvZFP30*, *VvZFP31*, *VvZFP32*, *VvZFP38*, and *VvZFP42*) reached their peaks. The significantly upregulated genes at 24 h were *VvZFP5*, *VvZFP6*, *VvZFP7*, *VvZFP12*, *VvZFP13*, *VvZFP14*, *VvZFP16*, *VvZFP22*-*VvZFP26*, *VvZFP29*-*VvZFP34*, *VvZFP36*, *VvZFP37*, *VvZFP40*, *VvZFP41*, *VvZFP43*, *VvZFP47*-*VvZFP50* and *VvZFP52*, of which *VvZFP43* had the highest relative expression level of 80.79, and the expression level of nine genes peaked at this treatment point. Thus, the majority of *VvZFPs* were induced to express by MejA but displayed with variant expression patterns.

#### 2.8.2. Expression of *VvZFP* Genes under ABA Treatment

After treatment with 100 mmol·L^−1^ ABA, a total of 7 *VvZFPs* were upregulated significantly at all time points (*p* < 0.05); no genes were completely downregulated at any time (Figure 9). For 3 h, the 35 genes with increased expression levels included *VvZFP1*-*VvZFP5*, *VvZFP8*, *VvZFP9*, *VvZFP11*, *VvZFP12*, *VvZFP14*-*VvZFP18*, *VvZFP20*-*VvZFP23*, *VvZFP25*-*VvZFP28*, *VvZFP33*, *VvZFP35*, *VvZFP37*, *VvZFP39*, *VvZFP43*, *VvZFP44*, *VvZFP48*, *VvZFP49*, and *VvZFP52* (*p* < 0.01), among which 15 genes peaked at this time point, and *VvZFP52* had the highest relative expression level of 224.92 (Figure 9). After treatment with 100 mmol·L^−1^ ABA for 6 h, the genes that were remarkably upregulated included *VvZFP1*-*VvZFP5*, *VvZFP8*, *VvZFP10*-*VvZFP28*, *VvZFP30*, *VvZFP35*, *VvZFP37-VvZFP40, VvZFP43*-*VvZFP47*, *VvZFP49*, and *VvZFP50*-*VvZFP52* (*p* < 0.01), with *VvZFP10* having the highest relative expression level of 6044.60, followed by *VvZFP46* and *VvZFP51* having the relative expression levels of 374.38 and 599.88, respectively; and there were 23 genes exhibiting peak expression at 6 h. After treatment with 100 mmol·L^−1^ ABA for 12 h, the genes with significantly increased expression levels were *VvZFP5*, *VvZFP6*, *VvZFP13*, *VvZFP23*, *VvZFP24*, *VvZFP29*-*VvZFP31*, *VvZFP36*, *VvZFP38*, *VvZFP41*, and *VvZFP43* (*p* < 0.01), among which *VvZFP13* and *VvZFP31* had the highest relative expression levels of 28.55 and 31.78, respectively; the expression level of *VvZFP6*, *VvZFP29*, *VvZFP30*, *VvZFP31*, and *VvZFP38* reached the peak at this time point. After treatment with 100 mmol·L^−1^ ABA for 24 h, the genes that were upregulated significantly included *VvZFP3*, *VvZFP6*, *VvZFP7*, *VvZFP11*, *VvZFP22*-*VvZFP25*, *VvZFP31-VvZFP34*, *VvZFP37*, *VvZFP40*-*VvZFP42*, *VvZFP47*, *VvZFP49*, and *VvZFP50* (*p* < 0.01), among which 8 genes reached their expression peaks, with *VvZFP32* exhibiting the highest relative expression level of 29.02. Thus, all members of the *VvZFP* family can be stimulated by ABA, yet they present diverse expression profiles; even when patterns appeared similarly, the expression levels varied.

#### 2.8.3. Expression of *VvZFP* Genes under SA Treatment

After treatment with 200 μmol·L^−1^ SA, the results of qRT-PCR revealed the expression of 8 *VvZFPs* was upregulated consistently across all time points (*p* < 0.05), only *VvZFP49* manifested a significant downregulation throughout the experiment (*p* < 0.05) (Figure 10). For 3 h, the remarkably upregulated genes were *VvZFP1-VvZFP6*, *VvZFP8*, *VvZFP9*, *VvZFP11*, *VvZFP12*, *VvZFP14*-*VvZFP17*, *VvZFP20*-*VvZFP23*, *VvZFP25*-*VvZFP28*, *VvZFP30*, *VvZFP33*-*VvZFP35*, *VvZFP37*, *VvZFP39*, *VvZFP44*, *VvZFP45*, *VvZFP48*, and *VvZFP51* (*p* < 0.01), of which 12 genes reached the peak of expression levels; furthermore, *VvZFP11* and *VvZFP39* had the highest relative expression levels of 76.89 and 80.53, respectively (Figure 10). With 200 μmol·L^−1^ SA treatment for 6 h, the upregulated genes were *VvZFP2*, *VvZFP4*, *VvZFP5*, *VvZFP10*-*VvZFP28*, *VvZFP30*, *VvZFP33*-*VvZFP35*, *VvZFP37*, *VvZFP39*, *VvZFP40*, *VvZFP42*, and *VvZFP44*-*VvZFP47* (*p* < 0.01), of which 23 genes reached the peak of expression, and *VvZFP10* had the highest relative expression level of 1870.33, followed by *VvZFP13*, *VvZFP46*, and *VvZFP47*, with the relative expression levels of 201.19, 143.23, and 234.68, respectively. With 200 μmol·L^−1^ SA treatment for 12 h, significantly upregulated expression genes were *VvZFP13*, *VvZFP19*, *VvZFP24, VvZFP25*, *VvZFP29*-*VvZFP31*, *VvZFP37*, *VvZFP38*, *VvZFP42*, and *VvZFP43* (*p* < 0.01), among which *VvZFP13* exhibited the highest relative expression level of 431.72, and *VvZFP6*, *VvZFP30-VvZFP32*, *VvZFP38*, and *VvZFP42* achieved their expression peaks. With 200 μmol·L^−1^ SA treatment for 24 h, the remarkably upregulated genes were *VvZFP1*-*VvZFP7*, *VvZFP16-VvZFP20*, *VvZFP22*, *VvZFP24*-*VvZFP26*, *VvZFP28*, *VvZFP29*, *VvZFP31*, *VvZFP37*, *VvZFP42*, *VvZFP43*, *VvZFP45, VvZFP48*, and *VvZFP50*-*VvZFP52*, among which *VvZFP24* exhibited the highest relative expression level of 21.47. These analyses indicated the expression of *VvZFPs* were induced by SA treatment, but with the differences in response speed and expression level.

#### 2.8.4. Expression of *VvZFP* Genes under 4 °C Treatment

Under the treatment of 4 °C, 6 of 52 *VvZFPs* were observed to be significantly upregulated across all time points (*p* < 0.05). In contrast, *VvZFP42* exhibited consistent downregulation throughout, while the expression level of *VvZFP29* had no changes during the treatment of 4 °C. The genes upregulated significantly after 4 °C treatment were *VvZFP1-VvZFP5*, *VvZFP7*-*VvZFP9*, *VvZFP11*-*VvZFP28*, *VvZFP30*, *VvZFP33*, *VvZFP35*, *VvZFP37*, *VvZFP39*, *VvZFP43*, *VvZFP44*, *VvZFP45*, *VvZFP47*-*VvZFP49*, and *VvZFP52* (*p* < 0.01), of which *VvZFP27* and *VvZFP52* showed the highest relative expression levels of 51.77 and 64.0, respectively (Figure 11), and 13 genes of them reached the expression peak. After 6 h of treatment at 4 °C, the remarkably upregulated genes were *VvZFP1-VvZFP5*, *VvZFP8*-*VvZFP28*, *VvZFP35-VvZFP37*, *VvZFP39*, *VvZFP40*, *VvZFP44-VvZFP47*, *VvZFP49*, *VvZFP50*-*VvZFP52* (*p* < 0.01), of which *VvZFP10* had the highest relative expression level of 5420.95, furthermore the expression levels of 29 upregulated genes peaked at 6 h. After 12 h of treatment at 4 °C, the genes with increased expression were *VvZFP5*, *VvZFP6*, *VvZFP7, VvZFP14*, *VvZFP22*-*VvZFP24*, *VvZFP26*, *VvZFP33*, *VvZFP34*, *VvZFP38*, *VvZFP41*, and *VvZFP47* (*p* < 0.01), of which *VvZFP6*, *VvZFP7*, *VvZFP24*, *VvZFP31*, *VvZFP34*, *VvZFP38*, and *VvZFP41* reached the peak expression, with *VvZFP7* exhibiting the highest relative expression level of 789.01. After 24 h of treatment at 4 °C, the genes that were upregulated significantly included *VvZFP5*, *VvZFP6*, *VvZFP22-VvZFP24*, *VvZFP29*, *VvZFP33*, *VvZFP34*.

*VvZFP41*, *VvZFP47*, *VvZFP50*, and *VvZFP51* (*p* < 0.01), of which *VvZFP51* exhibited the highest relative expression level of 8.27. These results showed majority of genes of *VvZFPs* respond to cold stress (4 °C), but the expression patterns were different.

### 2.9. Subcellular Localization of VvZFP10 and VvZFP46 Proteins

To determine the subcellular localization of the *VvZFP10* and *VvZFP46* proteins, Agrobacterium containing the recombinant plasmids pCAMBIA2300-VvZFP10-GFP and pCAMBIA2300-VvZFP46-GFP was injected into tobacco leaves for transient expression, with pCAMBIA2300-GFP as control. As depicted in Figure 12, *VvZFP10* and *VvZFP46* were localized in the nucleus of tobacco cells. Accordingly, we inferred that *VvZFP10* and VvZFP46 are located in the nucleus.

## 3. Discussion

### 3.1. Identification of Q-Type C2H2 ZFPs in Grapes

The Q-type ZFPs contain a domain consisting of approximately 25 amino acids and two conserved Cys and His residues, with a consensus sequence of CX2–4CX3FX3QALGGHX3–5H [29]. In total, 52 Q-type C2H2 *VvZFPs*, 54 C2H2 Q-type *VrZFPs*, and 55 Q-type C2H2 *VaZFPs* were identified in the genomes of *V. vinifera* PN40024, *V. riparia*, *V. amurensis*, respectively, and these genes have also been identified in other plants [23,33,34]. Analysis of the physicochemical properties of VvZFPs, VrZFPs, and VaZFPs revealed similarity between these genes; for example, the CDS lengths of *VvZFPs*, *VaZFPs* and *VrZFPs* were found to be in the range 468–1830 bp, 474–2583 bp, and 396–1836 bp, respectively, and their pI was found to be in the range 4.79–9.58, 4.94–10, and 4.78–9.3, respectively. VvZFP, VrZFP, and VaZFP proteins were predicted to be localized in the nucleus, except for VaZFP24 and VrZFP10, which were predicted to be localized in the cell membrane. To explore the subcellular localization of the VvZFP10 and VvZFP46 proteins, they were introduced into *Nicotiana benthamiana* leaves, which indicated that VvZFP10 and VvZFP46 were localized in the nucleus (Figure 12).

In this study, VvZFP, VrZFP, and VaZFP proteins were found to contain at least one C2H2 domain with a QALGGH motif, and Q-type VvZFP (VrZFP and VaZFP) proteins were found to have more than two zinc finger domains; these characteristics also have been reported in ZFPs in Arabidopsis and rice genomes [23,26]. A thorough examination of various Q-type C2H2 ZFP subfamily members possessing at least one QALGGH-containing C2H2 domain in Arabidopsis [23] revealed 57 members, which is higher than the number of *VvZFP*, *VrZFP*, and *VaZFP* members identified in this study. Most previously characterized *ZFPs* in plants reported to date are transcriptional repressors [18,35,36,37,38]. Interestingly, we found that the potential EAR motifs, which are associated with the ethylene-responsive element binding factor, are the predominant transcriptional repression motifs in plants, and “DLNxxP” is the most common type [39]. In this study, motif 6 contained “LDLNLRL”. Overall, 44 VvZFPs containing the EAR motif “LDLNLRL” were present at the C-terminus, whereas only 4 VvZFPs (VvZFP29, VvZFP30, VvZFP32 and VvZFP42) containing EAR motifs were present at the N-terminus (Figure 2), and 4 VvZFPs (VvZFP2, VvZFP22, VvZFP44, and VvZFP51) did not contain the EAR motif.

Based on the *VvZFP*, *VaZFP*, *VrZFP* gene structure and motif distribution, *ZFP* genes were divided into 8 groups according to phylogenetic analysis. Group 1 contained mainly two “QALGGH” motifs and one EAR motif. Group 2 contained the largest number and variety of motifs, mainly three “QALGGH” motifs and one EAR motif. Group 3 mainly contained three “QALGGH” motifs, one L-box motif, one B-box motif, and one EAR motif. Group 4 mainly contained two “QALGGH” motifs, one L-box motif, one B-box motif, and one EAR motif. Group 5 mainly contained one “QALGGH” motif and one EAR motif. Group 6 contained mainly one “QALGGH” motif. Group 7 contained mainly one “QALGGH” motif and one EAR motif. Group 8 mainly contained one “QALGGH” motif and one EAR motif. B-box and L-box were found to be located in the N-terminal region of some 2-fingered proteins, consistent with the results of a previous study [6]. B-box and L-box are conserved in some 2-fingered Q-type ZFP proteins, as reported by Sakamoto et al. [40]. Amino acid sequences in the zinc finger domain of these three (VvZFP, VaZFP, VrZFP) 1-fingered ZFP proteins are 100% identical. Thus, they compete for the same DNA-binding site and potentially regulate target gene expression by changing their expression under different physiological and environmental conditions. Two clade VI TaZFP proteins also share high sequence homology in zinc finger domains, as well as conserved EAR, B-box, and L-box motifs, with previously characterized ZFPs from other species that were reported to be upregulated under abiotic stress, namely Arabidopsis *AZF2* and *AtZat10* [15,40], Petunia *ZPT2-3* [20], and soybean SCOF-1 [17]. As observed in this study, *VvZFP10* exhibited a distinct response pattern to MeJA, ABA, SA, and 4 °C low temperatures stress compared with *VvZFP6* and *VvZFP7* (Figure 8, Figure 9, Figure 10 and Figure 11).

### 3.2. Chromosomal Localization, Gene Duplication, Ka/Ks, and Codon Usage Bias Analyses

This study identified 52 *VvZFP* genes unevenly distributed on 18 chromosomes, 55 *VaZFP* genes unevenly distributed on 17 chromosomes, and 54 *VrZFP* genes unevenly distributed on 17 chromosomes. Among these, *VvZFP* and *VaZFP* showed seven gene tandem duplications on chromosome 6, and *VrZFP* showed six gene tandem duplications on chromosome Chr1, suggesting that the *ZFP* genes among the three grapevine varieties share similar evolutionary characteristics and some selective variability (Figure 4A–C). In addition to tandem duplication, 19 segmental duplication events of *VvZFP*, 18 segmental duplication events of *VrZFP*, and nine segmental duplication events of *VaZFP* were observed using MCScanX methods (Figure 5A–C). The results of this study are consistent with those of a previous study [9]. It has been documented that gene duplication increases diversification relative to single gene-copy progenitors, leading to alterations in the genetic system and phenotype that are essential for evolutionary adaptation [41]. For example, the C-repeat binding factor (*CBF*) gene family based on gene duplication events expanded to six members in Arabidopsis; functional investigation of *CBF*1, 2, and 3, belonging to tandemly repeated genes, indicated that they were induced rapidly under low temperature [42], whereas *CBF4* is induced in response to drought stress, which can be attributed to the diversification of promoter regulatory elements [43]. In the present study, we found similar or differential expression patterns between tandem repeat gene pairs, for example, the *VvZFP12* and *VvZFP24* gene pairs exhibited similar expression patterns under MeJA, ABA, SA, and 4 °C low-temperature stress (*VvZFP12* was upregulated between 3 and 6 h of treatment, and *VvZFP24* was upregulated between 12 and 24 h of treatment), whereas the *VvZFP49* and *VvZFP27* gene pairs exhibited differential expression patterns (*VvZFP49* and *VvZFP27* were upregulated at 3 h and 6 h when subjected to MeJA, ABA, and 4 °C cold stress treatments; under SA treatment, *VvZFP49* was downregulated at 3 and 6 h, while *VvZFP27* was upregulated). These results clearly indicate that evolutionary changes in the genome promote genetic diversity, flexibility, and adaptability and cause alterations in gene expression.

The rate of nonsynonymous (Ka) and synonymous (Ks) substitutions is the basis for evaluating the positive selection pressure of duplication events. Ka/Ks = 1 indicates neutral selection, Ka/Ks < 1 denotes purification selection, and Ka/Ks > 1 signifies positive selection [9]. In *V. vinifera*, 293 gene pairs exhibited Ka/Ks > 1, two gene pairs had Ka/Ks = 1, and 344 gene pairs had Ka/Ks < 1. In *V. riparia*, 350 gene pairs had Ka/Ks > 1, three gene pairs exhibited Ka/Ks = 1, and 189 gene pairs exhibited Ka/Ks < 1. In *V. amurensis*, 418 gene pairs had Ka/Ks > 1, seven gene pairs had Ka/Ks = 1, and 234 gene pairs had Ka/Ks < 1. These results showed that during evolution, the Q-type C2H2 *ZFPs* were dominated by purification selection in *V. vinifera* but by positive selection in *V. amurensis* and *V. riparia* [9]. RSCU is defined as the ratio of the observed frequency to the expected frequency of codons when all the synonymous codons for those amino acids are used equally [44]. RSCU > 1 indicates that the corresponding codon is used more frequently than expected, whereas RSCU < 1 indicates that the codon is used less frequently than expected [45]. RSCU of the *VvZFPs*, *VrZFPs*, and *VaZFPs* was determined by averaging the corresponding RSCU values of amino acids (Appendix A). *VvZFP*, *VrZFP*, and *VaZFP* family members exhibit the preferential bias for A, U, and C at the third codon position, for example, Phe prefers codon UUC; Gln prefers CAA; Cys prefers UGC; Ser prefers UCU, UCC, and UCA; and Gly prefers GGU and GGA.

### 3.3. Responses to MeJA, ABA, SA, and Low-Temperature (4 °C) Treatments

Q-type C2H2 *ZFPs* have been reported to play a role in plant response to abiotic stresses [46,47]. Q-type *ZFPs*, *TaZFP1B*, and *TaZFP2D* are associated with aluminum tolerance and upregulated by H_2_O_2_ treatment in root tips [48]. Seven *TaZFP* genes upregulated under drought stress showed significant upregulation after ABA treatment [6]. The mRNA levels of eight genes, which were induced in response to drought in the leaves, remained stable throughout the 26 h ABA treatment [6]. Recently, the gene STZ/ZAT10 of A. thaliana was reported to be involved in the jasmonic signaling pathway [49]. MeJA treatment for 30 min had a slight effect on *PtaZFP2* mRNA accumulation [50]. To verify whether the Q-type C2H2 *ZFPs* respond to exogenous hormones, in this study, we designed experiments and validated the results by determining the relative gene expression. Following the treatment with 100 μmol·L^−1^ MeJA for 3 h, 6 h, 12 h, and 24 h, *VvZFP52*, *VvZFP10*, *VvZFP38*, and *VvZFP43* exhibited the highest relative expression levels of 58.94, 6824.21, 372.04, and 80.79, respectively, and reached their expression peaks at different time points. After treatment with 100 mmol·L^−1^ ABA for 3 h, *VvZFP52* had the highest relative expression level of 224.92; Under ABA treatment for 6 h, *VvZFP10* had the highest relative expression level of 6044.60, followed by *VvZFP46* and *VvZFP51*, with the expression levels of 374.38 and 599.88, respectively. Under ABA treatment for 12 h, *VvZFP31* had the highest relative expression levels of 31.78, respectively, whereas for 24 h, *VvZFP32* exhibited the highest relative expression level of 29.02. Similarly, under 200 μmol·L^−1^ SA treatment for 3 h, *VvZFP11* and *VvZFP39* had the highest relative expression levels of 76.89 and 80.53, respectively; under 6 h SA treatment, *VvZFP10* had the highest relative expression level of 1870.33, followed by *VvZFP13*, *VvZFP46*, *VvZFP47*, with relative expression levels of 201.19, 143.23, and 234.68, respectively; under 12 h SA treatment, *VvZFP13* had the highest relative expression level of 431.72; and under SA treatment for 24 h, *VvZFP24* exhibited the highest relative expression level of 21.47. These experimental results showed that Q-type C2H2 *ZFPs* in grapevine are differentially expressed at different hormone treatment stages and time points, suggesting that these genes exhibit different expression patterns in response to exogenous hormone treatments.

It has been demonstrated that *TaZFP* may have an important role in abiotic stress responses in wheat, as observed in other plants [46,47,51]. A study characterized the role of *PtaZFP* in poplar by analyzing the expression of two-fingered C2H2 *ZFPs* in different organs (leaf laminae, stems, and roots) and in response to cold stress (4 °C) [50]. *SlCZFP1* was reported to promote cold tolerance in *Solanum lycopersicum* [52]. We experimentally verified whether the Q-type C2H2 *ZFPs* respond to 4 °C low-temperature stress and validated the results by determining the relative expression of the genes. Under 4 °C low-temperature stress for 3 h, *VvZFP27* and *VvZFP52* exhibited the highest relative expression levels of 51.77 and 64.0, respectively; after treatment for 6 h, *VvZFP10* had the highest relative expression level of 5420.95; after treatment for 12 h, *VvZFP7* had the highest relative expression level of 789.01; and after treatment for 24 h, *VvZFP51* had the highest relative expression level of 8.27. These results suggest that the cis-acting elements of both *VvZFP7* and *VvZFP10* contain LTR elements. Our experimental results further indicated that Q-type C2H2 *ZFPs* in grapes are differentially expressed under low-temperature (4 °C) stress at different time points, suggesting that these genes exhibit differential expression patterns in response to low-temperature (4 °C) stress.

## 4. Materials and Methods

### 4.1. Identification of the VvZFP, VrZFP, and VaZFP Genes

The genome sequence information and gene annotation file (general feature format, GFF) of ‘Pinot noir’ (*Vitis vinifera* L.) were downloaded from Ensembl plants database (https://plants.ensembl.org/index.html, accessed on 18 April 2022). Whole genome information of ‘Shanputao’ (*V. amurensis*) was downloaded from the VITSGDB database (http://vitisgdb.ynau.edu.cn/downloads.html, accessed on 5 May 2021) [30], and the genome information of *V. riparia* was obtained from the NCBI database [53]. Fifty-seven ZFP amino acid sequences of *Arabidopsis thaliana* were obtained from the TAIR database (https://www.rabidopsis.org/, accessed on 23 August 2022).

To ensure the credibility of the results, two strategies were employed while identifying the ZFP gene family members of the grapevine. First, based on the hidden Markov model of the ZFP protein domain (PF00096), which was downloaded from the Pfam database (http://pfam.xfam.org/, accessed on 6 March 2021) as a query, the protein sequence data of the grapevine were searched using “hmmsearch” of HMMER 3.1 software. Second, a local BLAST database was generated using the whole genome protein sequences of grapevine, and 57 ZFP protein sequences of Arabidopsis were used as seed sequences to perform the local BLAST search with an E-value ≤ 1 × 10^−5^ [54]. The results obtained using the two methods were combined to remove redundant sequences. For conserved domain validation, the candidate ZFP protein sequences were submitted to the conserved domain database (CDD, https://www.ncbi.nlm.nih.gov/Structure/cdd/cdds.html, accessed on 12 June 2022) on the NCBI website and SMART (http://Smart.embl-Heidelberg.de/, accessed on 20 June 2022). Finally, manual screening was further performed to identify ZFP family members, and the proteins lacking specific structural domains of ZFP were removed.

### 4.2. Physicochemical Property and Subcellular Localization

The Compute pI/Mw tool on the online website Expasy (https://web.expasy.org/compute_pi/, accessed on 8 August 2022) was used to predict the isoelectric points (pI) and molecular weights of the VvZFP, VrZFP, VaZFP family members, and their subcellular localization was predicted using WoLF PSORT (https://www.expasy.org/compute_pi/ wolfpsort.hgc.jp/, accessed on 1 September 2022).

### 4.3. Analysis of Phylogenetic Clustering, Cis-Elements, and Protein Conserved Motifs

Multiple alignment of the protein sequences of *V. vinifera*, *V. riparia*, *V. amurensis*, and Arabidopsis was performed using ClustalW in MEGA7.0 [28], and a phylogenetic tree was constructed using the neighbor-joining (NJ) method. The method was executed using the following parameters: Jones–Talor–Thornton model, complete deletion, and 1000 repetitions.

To search for plant promoter cis-acting elements, the 2000-bp genome sequence upstream of the initiation codon of each *VvZFP*, *VrZFP*, and *VaZFP* gene was extracted from the grape whole genome database. The online database PlantCARE (http://bioinformatics.psb.ugent.be/webtools/plantcare/html/, accessed on 10 September 2022) was used to predict cis-acting elements, and the data were visualized using Box Plot tools in Hiplot Pro (https://hiplot.com.cn/, accessed on 2 November 2022), which is a comprehensive web service for biomedical data analysis and visualization.

The online software Multiple Em for Motif Elicitation (MEME Suite 5.5.4) was used to identify the conserved motifs (motifs) in VvZFP, VrZFP, and VaZFP proteins; the maximum value of motifs was set to 15, whereas those of other parameters were set as default [55]. Finally, TBtools 1.118.0.0 software was used to visualize the conserved motifs of these genes [56].

### 4.4. Chromosomal Location Analysis of the VvZFP, VrZFP, VaZFP Genes

The Gene Location Visualize tool in TBtools 1.118.0.0 software was employed to generate chromosome-based localization plots for the genes [56], with grape genome data files and gene ID information as materials.

### 4.5. Codon Usage Bias Analysis

Codon usage bias refers to the uneven utilization of synonymous codons to encode for an amino acid. The characteristics of codon usage were examined by using coding sequences of the *ZFP* genes as the subject [57,58]. This analysis encompassed various aspects, such as the evaluation of RSCU, determination of the codon adaptation index (CAI) and codon bias index (CBI), assessment of the frequency of optimal codons (FOPs), and examination of GC3s and GCs. All analyses were performed using the web-based program CodonW 1.4.2. A heat map of RSCU values was drawn using TBtools 1.118.0.0 [36], draw parameters were set to the cluster method (Complete) and dist method (Euclidean). Furthermore, correlation analyses were conducted to explore the relationships between codon usage bias and parameters including T3s, C3s, A3s, G3s, GC, GC3s, CBI, FOPs, Nc, L_sym, L_aa, Gravy and Aromo. 

### 4.6. Synteny and Evolution Selection Pressure Analyses

To analyze the synteny among different *ZFP* genes, we obtained the CDS.fasta files and transcripts.gff (or gff3) files for apple (*Malus domestic*), peach (*Prunus persica*), and *Arabidopsis* from the TIAR database. Using TBtools synteny, we identified the *ZFP* gene pairs and obtained the resulting diagram [56,59]. For duplicate gene pairs or triplicate gene groups (between any two genes within a triplicate gene group), we calculated the nonsynonymous substitution rate/synonymous substitution rate (Ka/Ks) values using the DnaSP 6.0, application developed by University of Barcelona. 

### 4.7. Organization and qRT-PCR Expression Analyses of VvZFP Family Genes

Grapevine tissue-specific expression data were obtained using gene microarray data of the gene expression omnibus (GEO) database on the NCBI website (NCBI accession number: GSE36128), which comprises data from 54 tissue samples of grapevine at different developmental stages, including those of roots, stems, leaves, tendrils, floral organs, pulp, pericarp, and seeds [31]. All *VvZFP* gene expression data were extracted according to their homologous gene ID number; finally, data on the expression of 41 *VvZFP* genes were obtained. The FPKM value was normalized to Log2 (FPKM+1) for data processing, and a heat map of gene expression was drawn using TBtools 1.118.0.0 [56], draw parameters were set to cluster method (Complete) and dist method (Euclidean).

Grapevine suspension cells from ‘Pinot Noir’ were cultured in Gamborg B5 [60] liquid medium (consisting of 20 g·L^−1^ sucrose, 0.1 mg·L^−1^ naphthalene acetic acid, and 0.2 mg·L^−1^ kinetin; pH 6.0) in the Cell Culture Laboratory of Gansu Agricultural University. The grapevine suspension cells were collected through funnel filtration after seven days of culture, and 4 g of the cells were weighed and added to 20 mL of B5 liquid medium supplemented with abscisic acid (ABA) (100 mmol·L^−1^), MeJA (100 µmol·L^−1^), and SA (200 µmol·L^−1^). The mixture was also subjected to low-temperature (4 °C) treatment. ABA, MeJA, SA, and low-temperature treatments were given for 3, 6, 12, and 24 h by incubating the suspension cells on a shaker at 110 r/min and 25 °C in the dark. Each treatment was performed in three biological replicates. Untreated suspension cells at each time point were used as the control, and cell samples in each group were collected according to the treatment time and stored at −80 °C.

The total RNA of the preserved grapevine cells was extracted using the TRIzol Kit (TaKaRa, Dalian, China) following the manufacturer’s instructions. cDNA was synthesized using the PrimeScript™ RT reagent Kit with gDNA Eraser (TaKaRa, Dalian, China). The SYBR^®^ Premix Ex Taq ™ II (TaKaRa, Dalian, China) was used for quantitative PCR (qPCR) amplification. The primers of 52 *VvZFP* genes were designed and synthesized using online software by Shanghai Sangon Biotechnology Co. Ltd. (Appendix A). The expression levels of *VvZFP* were normalized with elongation factor-1α (*EF-1α*, VIT_206S0004g03240) through real-time PCR by using the Light Cycler^®^ 96 real-time PCR system (Roche, Basel, Switzerland). qRT-PCR was performed in the following steps: denaturation at 95 °C for 30 s; amplification in a two-step procedure: denaturation for 15 s, and 40 cycles of annealing and extension for 60 s at 60 °C. The relative expression of the genes was calculated using 2^−∆∆Ct^ method [38]. Statistical analysis was performed using SPSS version 26.0, pairwise comparison using the LSD method. The plot of gene expression was generated using R software (v.4.2.2) package “ggpubr” (v0.4.0) [61] and “ggplot2” (v3.4.2) [62] through Hiplot Pro (https://hiplot.com.cn/, accessed on 3 April 2023), a comprehensive web service for biomedical data analysis and visualization.

### 4.8. Cloning and Subcellular Localization Analysis of VvZFP10 and VvZFP46

Primer sequences for cloning *VvZFP10* were as follows: ZFP10-F: 5′-AGAACACGG GGGACGAGCTCATGGAACAACTAAGGAAGGAGCCAT-3′ and ZFP10-R 5′-ACATG GTGTCGACTCTAGAGAGCTTGAGAG ACAAGTCAAGTTTC-3′; primers for *VvZFP46* were: ZFP46-F: 5′-AGAACACGGGGGACGAGCTCATGGAGAAGCACAAGTGCAAG CTG-3′ and ZFP46-R 5′-ACCATGGTGTCGACTCTGATCGTCTGATGGGGTTCACGAA C-3′. The amplification reaction included a denaturation step at 98 °C for 3 min, followed by 35 cycles of annealing for 10 s at 98 °C and for 30 s at 68 °C, and final extension at 72 °C for 5 min. PCR products were recovered using TaKaRa Gel Recovery Kit (TaKaRa, Dalian, China) and ligated into a PCAMBIA2300-GFP vector. Finally, the vector was transformed into Escherichia coli DH5α, and sequencing was performed after amplification by Shanghai Sangon Biotechnology Co. Ltd.

To determine the subcellular localization of *VvZFP10* and *VvZFP46*, the successfully sequenced plasmids of the above-mentioned vector for pCAMBIA2300-VvZFP10-GFP and pCAMBIA2300-VvZFP46-GFP were transferred into Agrobacterium tumefaciens GV1303. Then, the bacterial suspension was injected into the abaxial surface of *Nicotiana benthamiana* leaves, which were incubated in the dark for 24 h and then transferred to an incubator for 2 days. The fluorescence signal was observed using laser confocal microscopy.

## 5. Conclusions

In this study, a total of 52 Q-type *VvZFPs*, 54 Q-type *VrZFPs*, and 55 Q-type *VaZFPs* were identified in the three grapevine genomes, respectively, and found that there was a difference in the physical and chemical properties, chromosome localization, gene structure, conserved motif and cis-acting elements of promoter among *VvZFPs*, *VrZFPs* and *VaZFPs*. Additionally, the collinearity analysis revealed 19 *VvZFP* gene pairs, 18 *VrZFP* gene pairs, and 9 *VaZFP* gene pairs. Further, we performed the interspecies collinearity analysis between the *ZFPs* of grape and the other five species, which revealed 326 pairs of collinear genes. These pairs of homologous genes are functionally similar, and the expansion of *ZFP* in grapevine was dominated by segment duplication and tandem duplication events. Finally, the expression patterns of *VvZFPs* under MeJA, ABA, SA and 4 °C revealed that Q-type C2H2 ZFP-encoding genes in grapevine involved in response to exogenous hormones and low-temperature tolerance, which lays the foundation for further study of grapevine *ZFP* gene function and the development of resistant germplasm resources in the future.

## Figures and Tables

**Figure 1 ijms-24-15180-f001:**
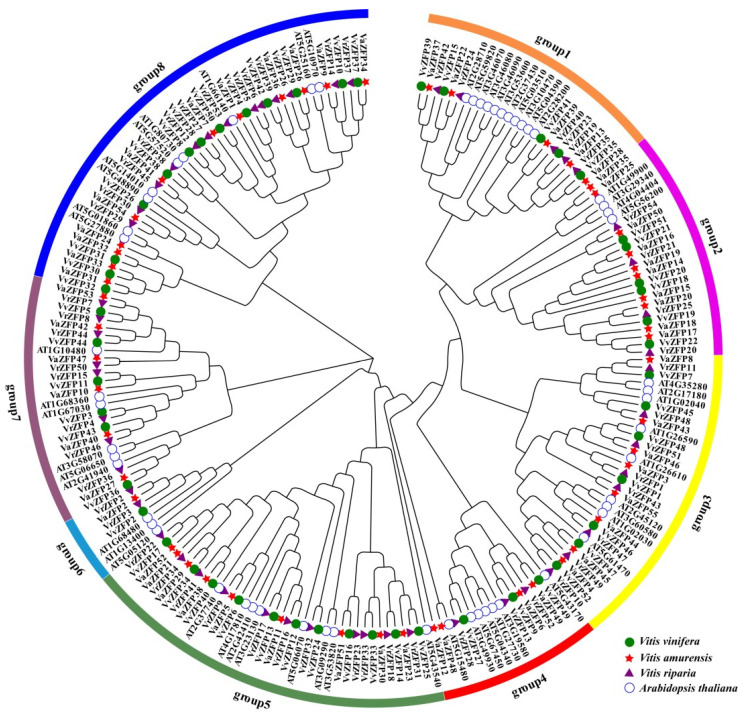
Phylogenetic analysis of ZFP Proteins in *Arabidopsis thaliana* (*At*), *Vitis vinifera* (*Vv*), *Vitis amurensis* (*Va*), and *Vitis riparia* (*Vr*). Note: The amino acid sequences of ZFP proteins were aligned using ClustalX, and a phylogenetic tree was constructed using the neighbor-joining method (NJ) method in MEGA 7.0 [28]. Each node is represented by a number that indicates the bootstrap value for 1000 replicates. The subgroups are marked by a colorful background.

**Figure 2 ijms-24-15180-f002:**
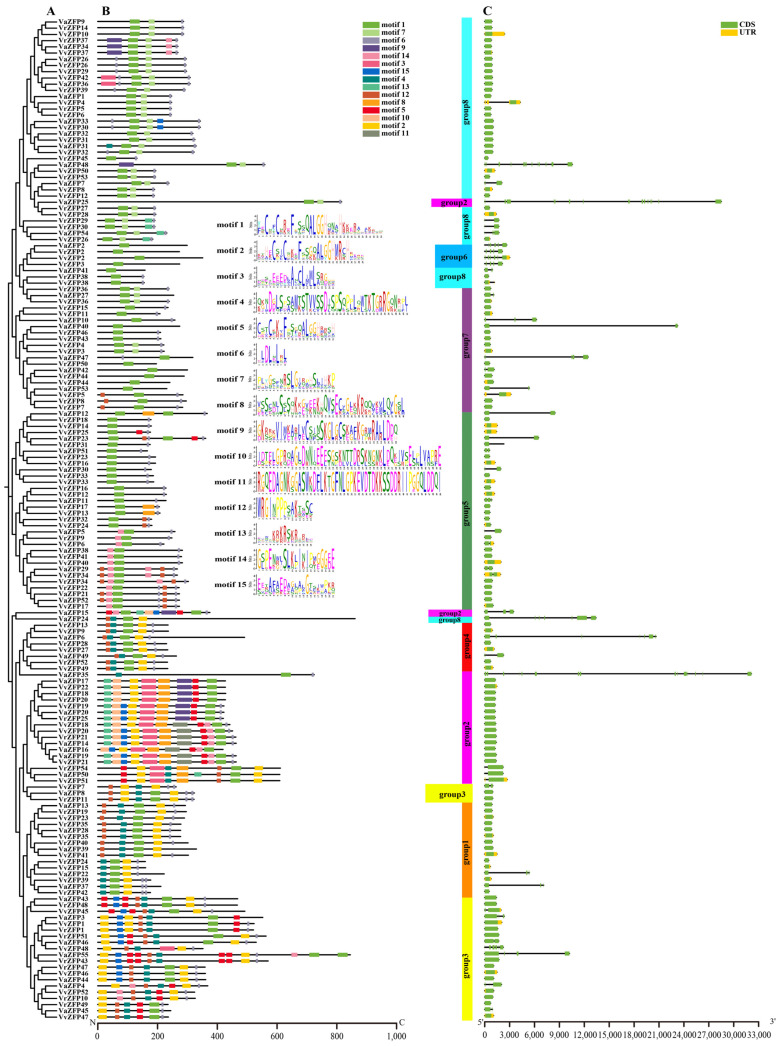
Phylogenetic relationships, gene structure, and architecture of conserved protein motifs in VvZFPs, VaZFPs, and VrZFPs. (**A**) The phylogenetic tree was constructed based on the protein sequences of ZFPs using MEGA 7.0 software with NJ method, bootstrap replicated 1000 times [28], Details of clusters are shown in different colors. (**B**) The motif composition of ZFP members; the motifs, numbered 1–15, are displayed in different colored boxes. (**C**) Exon–intron structure of *ZFPs*, Yellow boxes indicate untranslated 5′- and 3′-regions; green boxes indicate exons; and green lines indicate introns.

**Figure 3 ijms-24-15180-f003:**
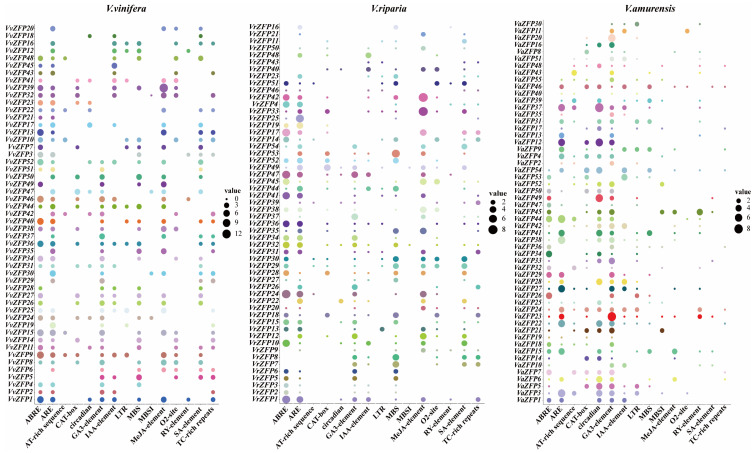
Distribution of major stress- and hormone response-related cis-acting regulatory elements in the promoters of *VvZFP*, *VrZFP*, and *VaZFP* genes. Note: Different colors represent different genes, the location of the circle corresponds to the corresponding cis-acting element, and the size of the circle represents the number of corresponding cis-elements.

**Figure 4 ijms-24-15180-f004:**
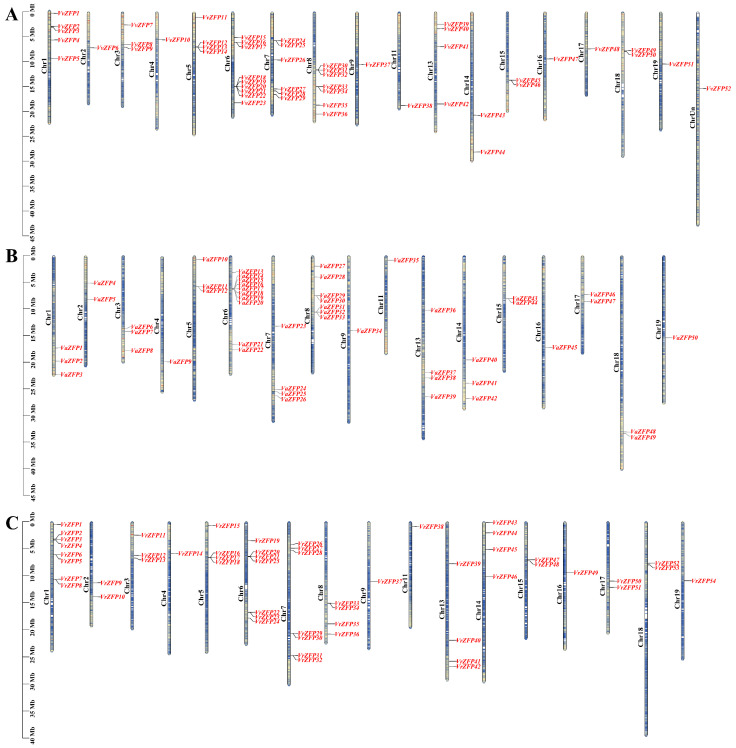
Chromosomal distributions of *ZFP* family genes in *Vitis vinifera*, *Vitis amurensis*, and *Vitis riparia*. Note: The physical positions of *ZFP* were mapped according to the grape genome, the left scale for the chromosomal length is mega bases (Mb), and the chromosome number is shown at the right of each chromosome. The color on the chromosome indicates the density of genes, with red indicating the highest density and blue indicating the lowest density. (**A**) Chromosome distribution of *VvZFPs*, (**B**) chromosome distribution of *VaZFPs*, (**C**) chromosome distribution of *VrZFPs*.

**Figure 5 ijms-24-15180-f005:**
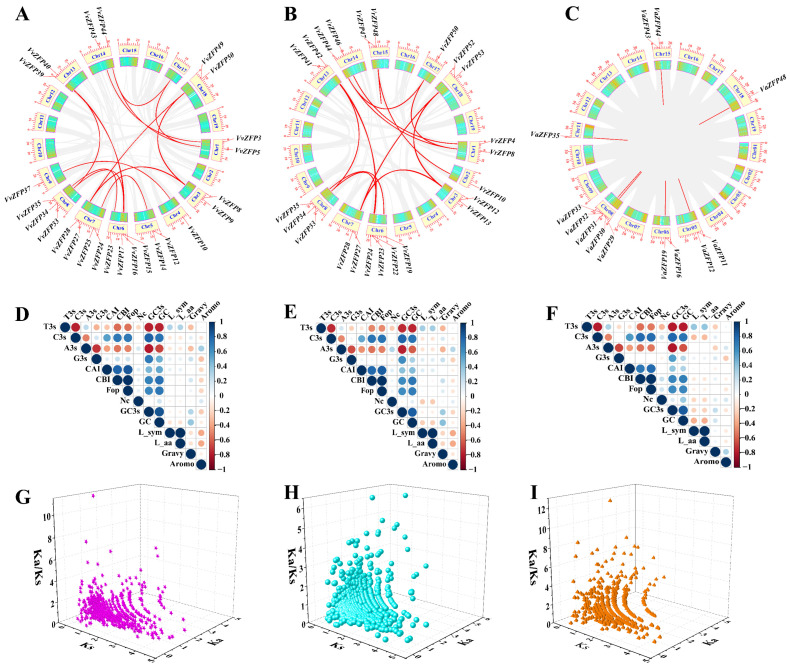
Schematics for the chromosomal distribution, interchromosomal relationships, Ka/Ks, and codon usage bias analysis of *ZFP* genes in the three grape species genomes. Note: (**A**–**C**), Syntenic relationship of *ZFP* in *V. vinifera*, *V. riparia*, and *V. amurensis*, respectively, genes in gray lines indicate all synteny blocks in the grape genome, and the red lines indicate duplicated *VvZFP*, *VrZFP*, *VaZFP* gene pairs; the chromosome number is indicated at the bottom of each chromosome. (**D**–**F**), Codon usage indexes correlation analysis of *VvZFPs*, *VrZFPs*, and *VaZFPs*, respectively; blue represents positive correlation, red represents negative correlation, and white represents no correlation; the larger the circle and the darker the color, the stronger the correlation and vice versa; “T3s, C3s, A3s, G3s”: refer to the T, C, A, and G content of the third position of the synonymous codon, respectively; “CAI” refers to the codon adaptation index; “CBI” refers to the codon bias index; “FOP” refers to the frequency of optimal codons; “GC3s” refers to the amount of the third codon (G+C); “GC” refers to the count of genes (G+C). “GC3s” refers to the amount of the third codon; “Nc”: refers to the number of codon; “L_sym” refer to the degeneracy of Symmetry codon; “L_aa”: refers to the number of synonymous and non-synonymous codons; “Gravy”: refers to hydrophobicity of protein; “Aromo”: refers to aromaticity of protein. (**G**–**I**), the Ka/Ks analysis of *VvZFPs*, *VrZFPs*, and *VaZFPs*, respectively; purple represents Ka/Ks of *VvZFPs*, blue represents Ka/Ks of *VrZFPs*, orange represents Ka/Ks of *VaZFPs*.

**Figure 6 ijms-24-15180-f006:**
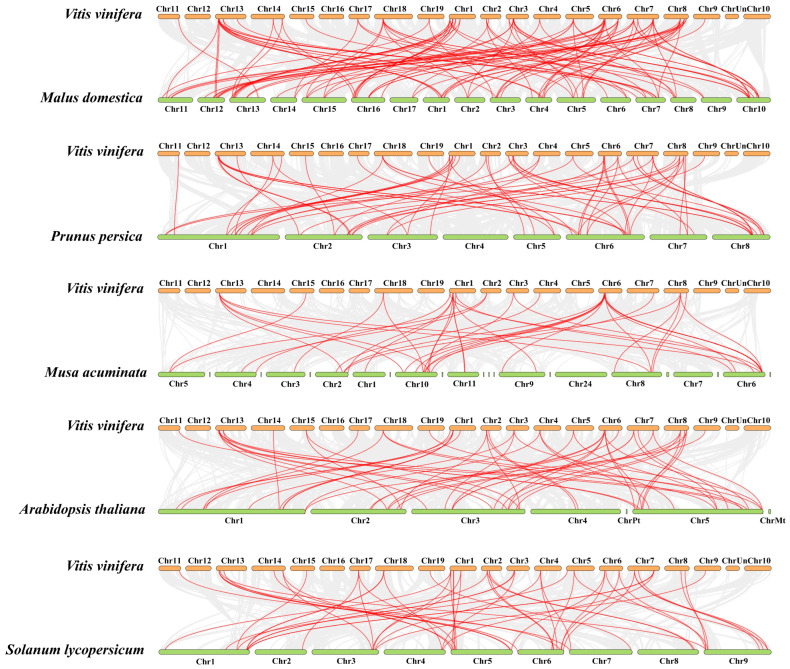
Synteny analysis of *VvZFP* genes of the five plant species. Note: Gray lines in the background indicate the collinear blocks within *Arabidopsis thaliana* and *Vitis vinifera*; *Malus domestica* and *Vitis vinifera*; *Prunus persica* and *Vitis vinifera*; *Musa acuminata* and *Vitis vinifera*; and *Solanum lycopersicum* and *Vitis vinifera* genomes, while the red lines highlight the syntenic *VvZFP* gene pairs among grape and other five plant species genomes.

**Figure 7 ijms-24-15180-f007:**
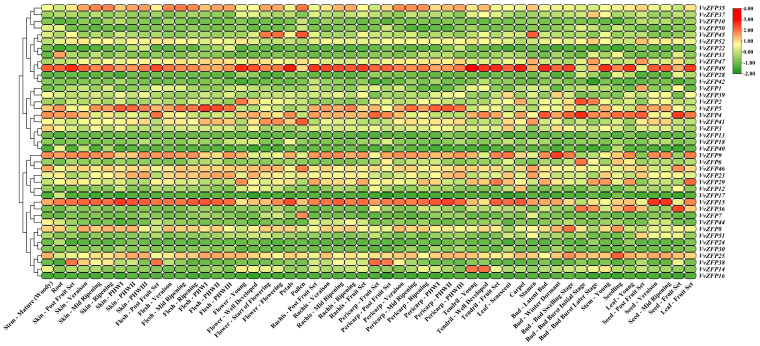
Hierarchical clustering of the expression profiles of 41 *VvZFP* genes in different tissues of grapevine at different growth and development stages. Note: Heatmaps were plotted using gene expression data of genechip microarrays, which were obtained from the GEO database on the NCBI website (NCBI accession number: GSE36128) [31]. The FPKM value was normalized to Log2 (FPKM+1) for data processing. Red and green shades represent the upregulated and downregulated expression levels, respectively. The scale denotes the relative expression level.

**Figure 8 ijms-24-15180-f008:**
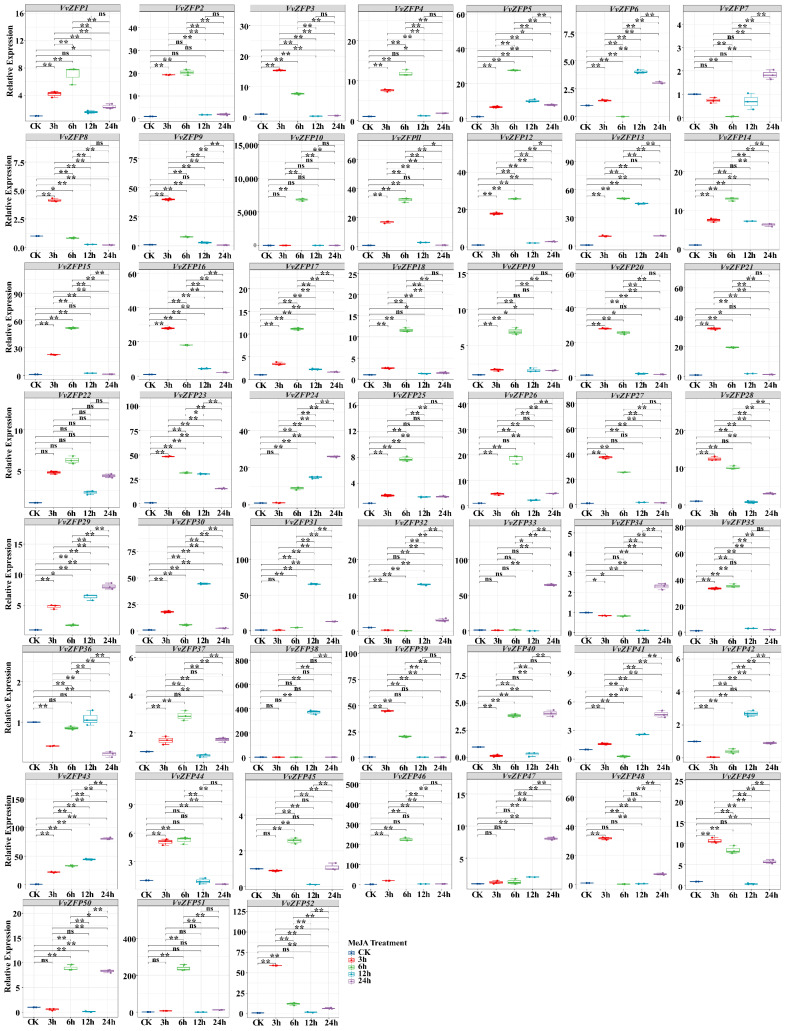
The expression of *VvZFP* genes under MeJA treatment. Note: The suspension cells samples were collected after 3 h, 6 h, 12 h and 24 h under 100 μM·L^−1^ MeJA. Untreated suspension cells at each time point were as control, respectively. The expression levels of *VvZFPs* were normalized with elongation factor-1α (*EF-1α*), and the relative expression was calculated using the 2^−∆∆Ct^ method [32]. Error bars represent the standard deviation for three biological replicates. Statistical analysis was performed using SPSS version 26.0, with pairwise comparison using the LSD method (‘*’ represents *p* < 0.05; ‘**’ represents *p* < 0.01).

**Figure 9 ijms-24-15180-f009:**
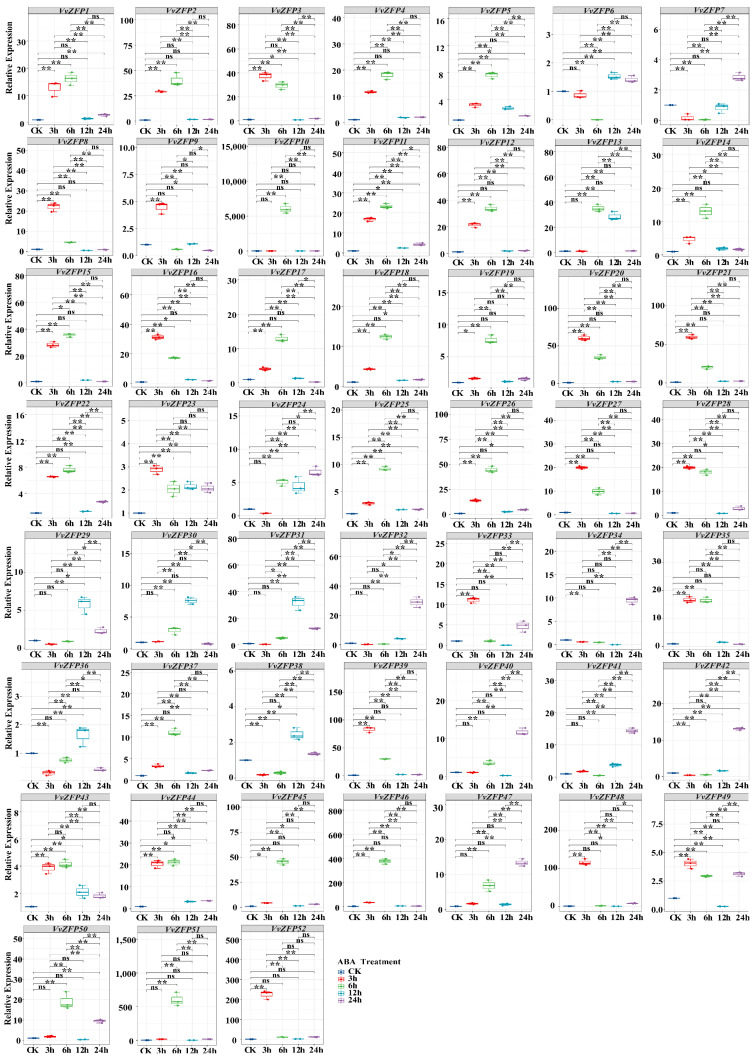
The expression of *VvZFP* genes under ABA treatment. Note: The suspension cells samples were collected after 3 h, 6 h, 12 h and 24 h under 100 μM·L^−1^ ABA. Untreated suspension cells at each time point were as control, respectively. The expression levels of *VvZFPs* were normalized with elongation factor-1α (*EF-1α*), and the relative expression was calculated using the 2^−∆∆Ct^ method [32]. Error bars represent the standard deviation for three biological replicates. Statistical analysis was performed using SPSS version 26.0, with pairwise comparison using the LSD method (‘*’ represents *p* < 0.05; ‘**’ represents *p* < 0.01).

**Figure 10 ijms-24-15180-f010:**
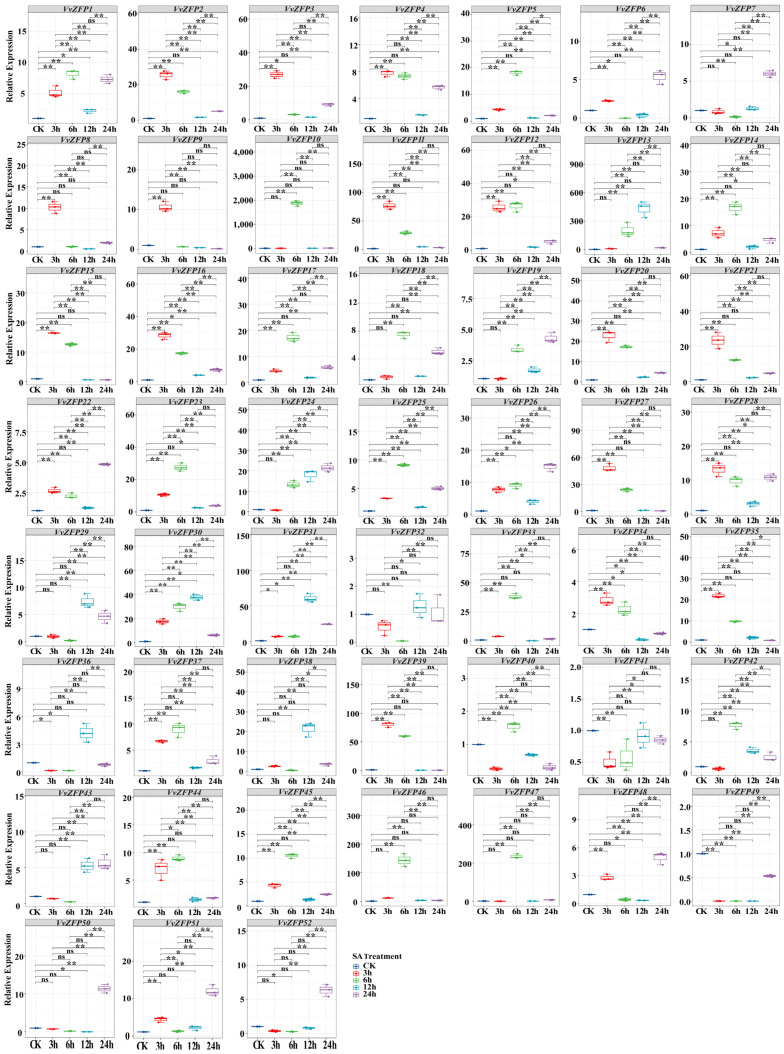
The expression of *VvZFP* genes under SA treatment. Notes: The suspension cells samples were collected after 3 h, 6 h, 12 h and 24 h under 200 μM·L^−1^ SA. Untreated suspension cells at each time point were as control, respectively. The expression levels of *VvZFPs* were normalized with elongation factor-1α (*EF-1α*), and the relative expression was calculated using the 2^−∆∆Ct^ method [32]. Error bars represent the standard deviation for three biological replicates. Statistical analysis was performed using SPSS version 26.0, with pairwise comparison using the LSD method (‘*’ represents *p* < 0.05; ‘**’ represents *p* < 0.01).

**Figure 11 ijms-24-15180-f011:**
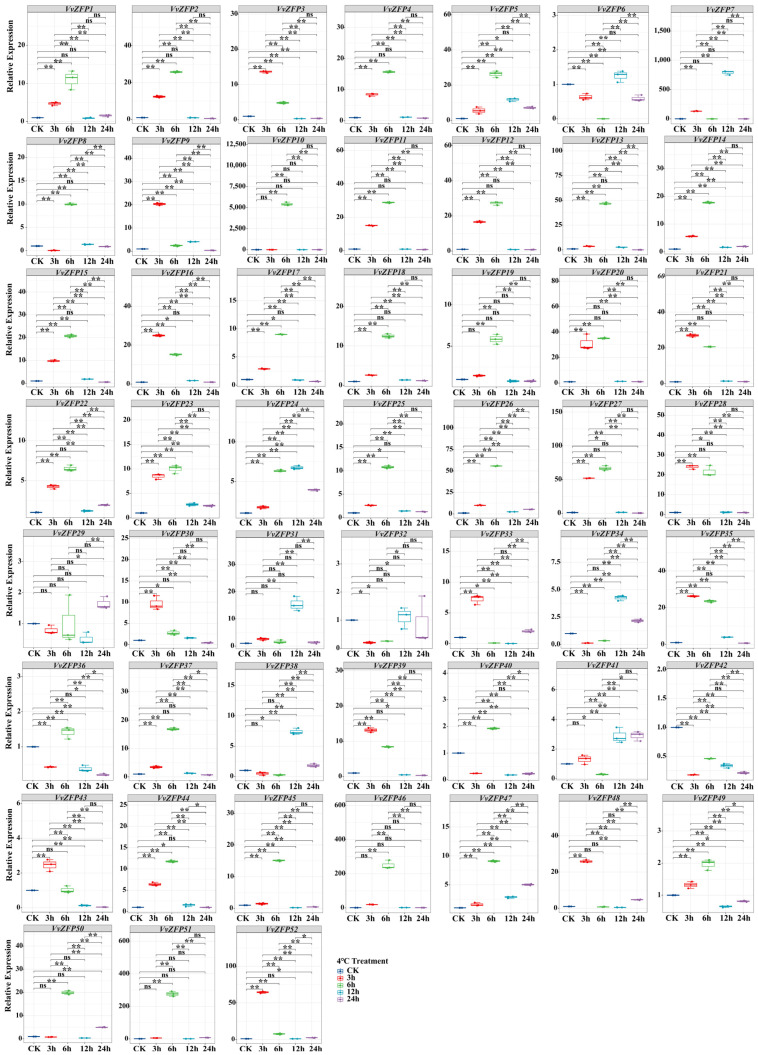
The expression of *VvZFP* genes under 4 °C treatment. Notes: The suspension cell samples were collected after 3 h, 6 h, 12 h and 24 h under 4 °C. Untreated suspension cells at each time point were as control, respectively. The expression levels of *VvZFPs* were normalized with elongation factor-1α (*EF-1α*), and the relative expression was calculated using the 2^−∆∆Ct^ method [32]. Error bars represent the standard deviation for three biological replicates. Statistical analysis was performed using SPSS version 26.0, with pairwise comparison using the LSD method (‘*’ represents *p* < 0.05; ‘**’ represents *p* < 0.01).

**Figure 12 ijms-24-15180-f012:**
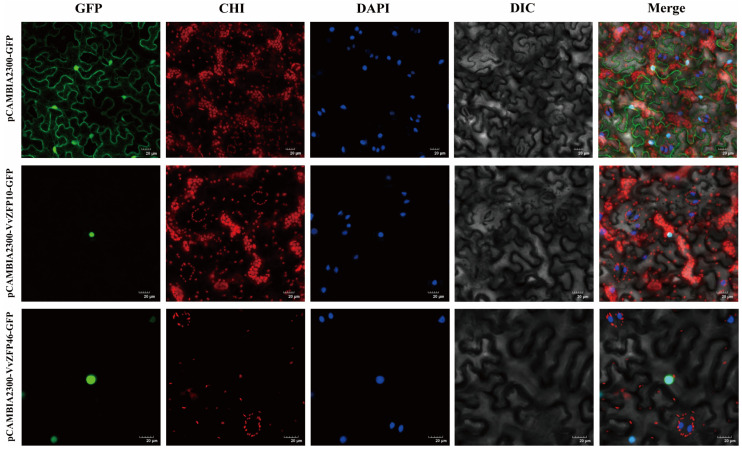
Subcellular localization of VvZFP10 and VvZFP46 proteins. Note: GFP stands for green fluorescence field, DAPI stands for DAPI field (cell nuclear staining), CHI stands for chloroplast autofluorescence field, DIC stands for bright field, and Merge stands for superimposed field. Excitation light wavelengths: GFP field: 488 nm, DAPI field: 358 nm, CHI field: 488 nm. Note: green fluorescence and chloroplast autofluorescence excitation light wavelengths were the same, and the acquisition light wavelengths were different. Bars = 20 μM.

**Table 1 ijms-24-15180-t001:** Physical and chemical properties of *ZFP* genes in grapes.

Gene Name	Gene ID	Chr: Start..End	CDS (bp)	Peptide (aa)	MW (Da)	pI	Aliphatic Index	Instability Index	Subcellular Localization	Alpha Helix (%)	Extended Strand (%)	Beta Turn (%)	Random Coil (%)
*VvZFP1*	Vitvi01g00063	Chr1: 616,107..617,675	1572	523	57,541.70	8.82	61.63	48.42	Nucleus.	32.88	17.37	0.00	49.75
*VvZFP2*	Vitvi01g01939	Chr1: 3,252,820..3,255,454	1056	351	37,407.67	7.55	59.03	50.27	Nucleus.	32.59	16.04	0.00	51.37
*VvZFP3*	Vitvi01g00309	Chr1: 3,422,695..3,423,360	669	222	23,139.71	8.37	56.24	53.3	Nucleus.	27.97	22.15	0.00	49.87
*VvZFP4*	Vitvi01g00529	Chr1: 6,009,188..6,009,928	744	247	26,776.90	5.43	65.85	53.07	Nucleus.	31.95	17.08	0.00	50.97
*VvZFP5*	Vitvi01g00845	Chr1: 9,724,651..9,725,541	894	297	33,581.02	9.10	58.54	64.60	Nucleus.	10.86	12.00	3.71	73.43
*VvZFP6*	Vitvi02g00686	Chr2: 7,241,834..7,242,586	666	221	24,181.37	5.06	55.09	61.21	Nucleus.	26.36	8.64	2.73	62.27
*VvZFP7*	Vitvi03g00230	Chr3: 2,660,982..2,661,950	789	262	28,337.08	8.42	69.96	58.16	Nucleus.	14.94	13.79	3.83	67.43
*VvZFP8*	Vitvi03g00582	Chr3: 6,548,735..6,549,304	573	190	21,159.71	6.52	57.83	50.21	Nucleus.	29.63	8.47	1.59	60.32
*VvZFP9*	Vitvi03g00630	Chr3: 7,125,512..7,126,219	711	236	25,656.66	8.53	52.38	70.23	Nucleus.	15.32	8.09	3.40	73.19
*VvZFP10*	Vitvi04g00527	Chr4: 5,630,158..5,631,018	864	287	31,544.12	5.96	65.56	56.93	Nucleus.	21.33	12.24	1.75	66.69
*VvZFP11*	Vitvi05g00082	Chr5: 1,128,866..1,129,489	627	208	22,072.40	8.59	47.25	77.98	Nucleus.	29.47	10.63	3.86	56.04
*VvZFP12*	Vitvi05g01913	Chr5: 7,082,504..7,083,190	690	229	25,470.51	8.72	63.68	53.86	Nucleus.	27.63	8.33	3.51	60.53
*VvZFP13*	Vitvi05g01914	Chr5: 7,094,682..7,095,305	627	208	23,008.67	6.21	64.59	66.8	Nucleus.	33.33	10.63	6.76	49.28
*VvZFP14*	Vitvi05g01915	Chr5: 7,102,567..7,103,103	540	179	20,359.12	6.97	34.61	62.89	Nucleus.	14.61	9.55	2.81	73.03
*VvZFP15*	Vitvi06g01682	Chr6: 5,152,508..5,152,987	483	160	17,811.68	9.15	81.51	46.90	Nucleus.	19.50	11.32	2.52	66.67
*VvZFP16*	Vitvi06g01710	Chr6: 6,156,545..6,157,123	582	193	20,968.39	6.90	57.50	38.51	Nucleus.	14.58	8.33	2.08	75.00
*VvZFP17*	Vitvi06g00521	Chr6: 6,160,463..6,161,281	822	273	30,657.63	9.19	66.11	63.46	Nucleus.	25.37	10.66	1.10	62.87
*VvZFP18*	Vitvi06g01862	Chr6: 14,981,645..14,982,970	1329	442	48,569.67	9.58	55.78	47.97	Nucleus.	24.49	8.16	3.17	64.17
*VvZFP19*	Vitvi06g01864	Chr6: 14,989,117..14,990,382	1269	422	45,557.15	9.08	51.48	60.45	Nucleus.	16.15	8.55	4.51	70.78
*VvZFP20*	Vitvi06g01865	Chr6: 15,002,349..15,003,701	1356	451	49,058.13	9.49	58.53	45.02	Nucleus.	18.67	10.00	3.78	65.56
*VvZFP21*	Vitvi06g01866	Chr6: 15,005,234..15,006,622	1392	463	50,164.48	9.31	59.76	46.43	Nucleus.	19.97	12.77	4.11	63.20
*VvZFP22*	Vitvi06g01867	Chr6: 15,013,314..15,014,594	1284	427	45,964.15	8.68	58.57	53.67	Nucleus.	23.00	8.22	3.76	65.02
*VvZFP23*	Vitvi06g01350	Chr6: 18,287,579..18,288,448	873	290	30,928.25	6.70	56.06	71.75	Nucleus.	10.73	11.07	3.46	74.74
*VvZFP24*	Vitvi07g02261	Chr7: 5,788,161..5,788,703	546	181	20,235.54	9.15	54.78	58.62	Nucleus.	26.67	10.00	3.89	59.44
*VvZFP25*	Vitvi07g02262	Chr7: 5,794,112..5,794,639	531	176	19,739.77	4.79	51.89	67.57	Nucleus.	22.29	10.86	2.86	64.00
*VvZFP26*	Vitvi07g02342	Chr7: 9,671,071..9,671,622	555	184	20,226.55	9.03	59.29	48.14	Nucleus.	24.59	11.48	3.28	60.66
*VvZFP27*	Vitvi07g01602	Chr7: 15,505,941..15,506,642	705	234	25,007.84	7.22	60.39	67.42	Nucleus.	16.31	6.44	3.00	74.25
*VvZFP28*	Vitvi07g01645	Chr7: 15,914,286..15,914,867	585	194	21,466.78	6.16	50.67	43.88	Nucleus.	30.57	9.84	1.55	58.03
*VvZFP29*	Vitvi07g01724	Chr7: 16,589,850..16,590,740	894	297	31,837.31	6.87	48.48	65.66	Nucleus.	12.50	21.96	7.43	58.11
*VvZFP30*	Vitvi08g04156	Chr8: 11,691,926..11,692,954	1032	343	37,210.24	9.37	51.46	61.82	Nucleus.	24.27	10.82	4.09	60.82
*VvZFP31*	Vitvi08g04157	Chr8: 11,698,586..11,699,560	978	325	35,886.02	8.11	59.01	54.74	Nucleus.	29.94	9.57	4.94	55.56
*VvZFP32*	Vitvi08g02113	Chr8: 11,710,332..11,711,297	969	322	34,967.55	8.99	49.84	45.38	Nucleus.	22.12	10.59	5.61	61.68
*VvZFP33*	Vitvi08g02203	Chr8: 15,040,331..15,040,891	1164	387	20,527.68	5.85	57.20	48.76	Nucleus.	15.59	15.05	2.15	67.20
*VvZFP34*	Vitvi08g01249	Chr8: 15,048,333..15,049,244	801	266	29,979.58	9.08	59.28	70.52	Nucleus.	24.91	8.30	2.64	64.15
*VvZFP35*	Vitvi08g01600	Chr8: 18,752,543..18,753,373	684	227	30,176.53	6.37	54.17	70.05	Nucleus.	11.96	13.41	3.26	71.38
*VvZFP36*	Vitvi08g01771	Chr8: 20,543,918..20,544,634	720	239	26,467.24	6.76	53.28	56.27	Nucleus.	15.97	13.03	1.26	69.75
*VvZFP37*	Vitvi09g00827	Chr9: 10,638,815..10,639,618	807	268	29,405.59	7.13	67.64	55.45	Nucleus.	19.10	13.11	2.25	65.54
*VvZFP38*	Vitvi11g01259	Chr11: 18,868,891..18,869,355	468	155	17,251.04	7.91	62.08	59.60	Nucleus.	27.92	14.29	3.25	54.55
*VvZFP39*	Vitvi13g00262	Chr13: 2,500,654..2,501,184	534	177	19,227.15	9.25	73.81	59.02	Nucleus.	22.16	14.77	1.70	61.36
*VvZFP40*	Vitvi13g00340	Chr13: 3,432,929..3,433,777	852	283	30,817.44	6.86	66.77	67.47	Nucleus.	26.24	8.51	2.13	63.12
*VvZFP41*	Vitvi13g00694	Chr13: 7,002,863..7,003,771	912	303	33,185.90	6.00	83.96	59.90	Nucleus.	16.89	14.24	4.30	64.57
*VvZFP42*	Vitvi13g01403	Chr13: 18,524,495..18,525,421	930	309	33,906.90	8.29	52.89	65.25	Nucleus.	21.1	10.06	2.27	66.56
*VvZFP43*	Vitvi14g02872	Chr14: 20,758,279..20,758,911	636	211	23,154.29	8.83	59.19	62.25	Nucleus.	24.76	13.33	0.48	61.43
*VvZFP44*	Vitvi14g01822	Chr14: 28,162,609..28,163,331	726	241	27,027.55	8.89	66.71	54.50	Nucleus.	30.83	11.67	3.75	53.75
*VvZFP45*	Vitvi15g00644	Chr15: 13,742,179..13,743,973	1479	492	53,927.39	6.94	61.81	53.23	Nucleus.	22.61	8.76	3.87	64.77
*VvZFP46*	Vitvi15g01472	Chr15: 13,755,848..13,756,927	1083	360	40,177.60	5.36	57.99	70.04	Nucleus.	30.08	9.47	3.06	57.38
*VvZFP47*	Vitvi16g00499	Chr16: 9,490,833..9,491,540	711	236	26,341.58	8.77	53.62	56.48	Nucleus.	20.43	12.77	3.83	62.98
*VvZFP48*	Vitvi17g00670	Chr17: 7,475,229..7,477,459	1059	352	38,525.41	9.42	61.14	57.18	Nucleus.	28.77	13.39	3.13	54.70
*VvZFP49*	Vitvi18g00675	Chr18: 7,732,710..7,733,411	705	234	25,291.37	8.91	61.24	72.28	Nucleus.	12.88	7.73	3.00	76.39
*VvZFP50*	Vitvi18g00708	Chr18: 8,000,360..8,000,941	585	194	21,621.29	8.37	53.11	54.30	Nucleus.	25.91	11.40	2.07	60.62
*VvZFP51*	Vitvi19g00912	Chr19: 10,503,115..10,505,322	1830	609	65,943.19	6.43	52.88	43.44	Nucleus.	19.74	9.87	0.00	70.39
*VvZFP52*	Vitvi02g00560	ChrUn: 15,415,314..15,416,285	975	324	35,711.51	5.80	55.91	70.98	Nucleus.	27.76	16.50	0.00	55.75
*VaZFP1*	VAG0100928.1	Chr1: 17,333,363..17,334,103	741	246	26,762.87	5.42	65.85	52.76	Nucleus	23.58	12.20	0.00	64.23
*VaZFP2*	VAG0101159.1	Chr1: 19,799,732..19,802,380	897	298	31,807.04	8.17	48.09	57.43	Nucleus	12.08	9.40	0.00	78.52
*VaZFP3*	VAG0101417.1	Chr1: 22,359,578..22,361,926	1656	551	61,040.87	9.00	62.29	48.67	Nucleus	26.13	13.79	0.00	60.07
*VaZFP4*	VAG0101986.1	Chr2: 5,166,395..5,168,415	1104	367	40,529.11	6.36	56.68	69.07	Nucleus	32.97	6.54	0.00	60.49
*VaZFP5*	VAG0102194.1	Chr2: 8,213,767..8,215,736	774	257	28,235.06	5.10	57.78	62.65	Nucleus	29.57	7.00	0.00	63.42
*VaZFP6*	VAG0103111.1	Chr3: 13,633,297..13,653,939	1473	490	54,091.12	8.10	72.63	53.50	Cell membrane, Nucleus	27.55	7.14	0.00	65.31
*VaZFP7*	VAG0103166.1	Chr3: 14,187,647..14,189,709	714	237	26,675.28	8.36	64.18	49.68	Nucleus	22.36	13.08	0.00	64.56
*VaZFP8*	VAG0103518.1	Chr3: 17,847,415..17,848,383	969	322	35,038.46	7.57	66.09	54.66	Nucleus	18.32	12.73	0.00	68.94
*VaZFP9*	VAG0104819.1	Chr4: 19,776,441..19,777,301	861	286	31,604.16	5.96	65.56	56.33	Nucleus	23.43	8.39	0.00	68.18
*VaZFP10*	VAG0105424.1	Chr5: 724,847..731,076	774	257	28,247.83	9.64	61.13	60.37	Nucleus	24.51	20.62	0.00	54.86
*VaZFP11*	VAG0105875.1	Chr5: 5,823,943..5,824,794	684	227	25,553.70	7.63	67.84	49.78	Nucleus	39.21	2.20	0.00	58.59
*VaZFP12*	VAG0105876.1	Chr5: 5,836,792..5,845,261	1098	365	40,943.95	8.24	43.10	64.52	Nucleus	19.45	3.84	0.00	76.71
*VaZFP13*	VAG0106999.1	Chr6: 3,224,617..3,225,495	879	292	31,485.92	6.27	55.48	69.89	Nucleus	7.88	15.41	0.00	76.71
*VaZFP14*	VAG0107168.1	Chr6: 6,152,873..6,154,261	1389	462	50,333.62	9.28	60.17	48.09	Nucleus	23.59	8.23	0.00	68.18
*VaZFP15*	VAG0107169.1	Chr6: 6,167,187..6,170,659	1125	374	40,810.87	9.39	58.21	51.67	Nucleus	17.38	13.10	0.00	69.52
*VaZFP16*	VAG0107172.1	Chr6: 6,206,628..6,207,884	1257	418	45,670.68	9.97	56.27	52.09	Nucleus	25.84	7.89	0.00	66.27
*VaZFP17*	VAG0107173.1	Chr6: 6,214,231..6,215,511	1281	426	45,912.17	8.66	60.63	53.82	Nucleus	8.45	13.62	0.00	77.93
*VaZFP18*	VAG0107180.1	Chr6: 6,339,579..6,340,862	1284	427	46,126.56	9.03	59.79	52.37	Nucleus	10.30	13.35	0.00	76.35
*VaZFP19*	VAG0107181.1	Chr6: 6,347,333..6,348,721	1389	462	50,062.39	9.44	59.76	46.49	Nucleus	21.86	8.23	0.00	69.91
*VaZFP20*	VAG0107185.1	Chr6: 6,369,265..6,370,530	1266	421	45,542.21	9.22	62.30	52.83	Nucleus	12.83	10.69	0.00	76.48
*VaZFP21*	VAG0107708.1	Chr6: 16,631,844..16,632,662	819	272	30,572.47	9.19	62.02	64.40	Nucleus	23.53	14.71	0.00	61.76
*VaZFP22*	VAG0107805.1	Chr6: 17,689,739..17,695,113	666	221	24,781.68	9.81	77.69	54.48	Nucleus	28.51	10.86	0.00	60.63
*VaZFP23*	VAG0109054.1	Chr7: 13,229,020..13,235,502	1083	360	40,449.95	5.69	56.14	59.44	Nucleus	23.33	10.56	0.00	66.11
*VaZFP24*	VAG0109724.1	Chr7: 25,148,488..25,161,891	2583	860	94,567.56	7.30	81.55	46.58	Cell membrane	36.51	13.26	0.00	50.23
*VaZFP25*	VAG0109762.1	Chr7: 25,543,220..25,571,722	2448	815	92,490.14	6.10	63.55	38.40	Nucleus	24.54	19.51	0.00	55.95
*VaZFP26*	VAG0109816.1	Chr7: 26,316,847..26,317,734	888	295	31,755.23	6.46	46. 00	66.23	Nucleus	15.93	13.22	0.00	70.85
*VaZFP27*	VAG0110325.1	Chr8: 1,994,129..1,995,217	762	253	28,208.24	8.26	51.66	52.61	Nucleus	13.83	15.81	0.00	70.36
*VaZFP28*	VAG0110521.1	Chr8: 4,033,984..4,034,814	831	276	30,176.53	6.37	54.17	70.05	Nucleus	28.39	7.26	0.00	64.35
*VaZFP29*	VAG0110874.1	Chr8: 7,544,736..7,545,647	798	265	29,979.58	9.08	59.28	70.52	Nucleus	17.84	7.60	0.00	74.56
*VaZFP30*	VAG0110875.1	Chr8: 7,551,607..7,553,542	537	178	19,482.45	6.30	53.82	54.81	Nucleus	23.03	6.18	0.00	70.79
*VaZFP31*	VAG0111156.1	Chr8: 10,604,900..10,605,886	987	328	35,580.28	8.88	48.78	46.77	Nucleus	12.32	15.58	0.00	72.10
*VaZFP32*	VAG0111157.1	Chr8: 10,616,669..10,617,622	954	317	35,110.09	8.39	59.09	50.64	Nucleus	28.39	7.26	0.00	64.35
*VaZFP33*	VAG0111158.1	Chr8: 10,623,270..10,624,298	1029	342	37,242.19	9.39	50.03	63.91	Nucleus	23.03	6.18	0.00	70.79
*VaZFP34*	VAG0112436.1	Chr9: 14,077,157..14,077,960	804	267	29,405.59	7.13	67.64	55.45	Nucleus	27.92	8.68	0.00	63.40
*VaZFP35*	VAG0114471.1	Chr11: 864,420..896,550	2172	723	80,321.46	4.94	75.17	42.25	Nucleus	15.55	7.93	0.00	76.52
*VaZFP36*	VAG0117437.1	Chr13: 10,279,271..10,280,197	927	308	33,975.97	8.30	52.89	65.28	Nucleus	28.37	7.09	0.00	64.54
*VaZFP37*	VAG0118243.1	Chr13: 2,1951,931..21,959,029	633	210	23,176.69	10.00	74.38	69.36	Nucleus	15.73	13.11	0.00	71.16
*VaZFP38*	VAG0118338.1	Chr13: 22,954,546..22,955,394	849	282	30,794.36	6.63	66.42	66.79	Nucleus	32.92	14.11	0.00	52.97
*VaZFP39*	VAG0118611.1	Chr13: 26,509,806..26,510,795	990	329	35,924.05	9.00	64.44	72.84	Nucleus	31.90	9.05	0.00	59.05
*VaZFP40*	VAG0119731.1	Chr14: 19,561,186..19,584,426	822	273	30,537.70	9.38	55.90	60.37	Nucleus	27.47	7.69	0.00	64.84
*VaZFP41*	VAG0120008.1	Chr14: 23,954,383..23,955,312	474	157	17,102.01	7.91	52.93	61.88	Nucleus	15.50	12.46	0.00	72.04
*VaZFP42*	VAG0120315.1	Chr14: 26,818,406..26,819,535	900	299	33,836.44	8.99	69.83	59.54	Nucleus	19.16	9.74	0.00	71.10
*VaZFP43*	VAG0121158.1	Chr15: 8,053,433..8,054,836	1404	467	51,114.12	6.76	60.60	47.41	Nucleus	22.06	12.63	0.00	65.31
*VaZFP44*	VAG0121159.1	Chr15: 8,066,755..8,067,837	1083	360	40,236.63	5.35	57.83	68.71	Nucleus	23.61	8.06	0.00	68.33
*VaZFP45*	VAG0122572.1	Chr16: 17,249,757..17,250,691	732	243	27,101.48	8.72	53.46	56.73	Nucleus	29.63	12.76	0.00	57.61
*VaZFP46*	VAG0123756.1	Chr17: 7,323,153..7,324,843	1590	529	58,536.29	8.57	58.66	54.59	Nucleus	24.01	15.12	0.00	60.87
*VaZFP47*	VAG0123852.1	Chr17: 8,584,588..8,597,041	954	317	35,447.08	6.92	65.27	49.71	Nucleus	30.28	7.89	0.00	61.83
*VaZFP48*	VAG0125708.1	Chr18: 33,103,767..33,114,322	1677	558	61,587.31	5.94	77.06	40.08	Nucleus	18.46	27.78	0.00	53.76
*VaZFP49*	VAG0125736.1	Chr18: 33,378,797..33,381,058	789	262	28,363.71	8.40	58.21	79.33	Nucleus	17.94	8.78	0.00	73.28
*VaZFP50*	VAG0126909.1	Chr19: 15,420,039..15,422,244	1827	608	66,030.26	6.36	53.36	44.38	Nucleus	19.57	10.53	0.00	69.90
*VaZFP51*	VAG0128236.1	Scaffold_187: 19,406..19,986	498	165	18,085.18	7.73	55.09	38.21	Nucleus	30.30	10.30	0.00	59.39
*VaZFP52*	VAG0128237.1	Scaffold_187: 23,941..24,759	819	272	30,553.43	9.08	62.02	65.27	Nucleus	23.90	15.07	0.00	61.03
*VaZFP53*	VAG0129335.1	Scaffold_503: 45,030..50,401	693	230	26,231.45	6.24	67.00	58.53	Nucleus	27.83	11.30	0.00	60.87
*VaZFP54*	VAG0130661.1	Scaffold_1,201: 6,417..8,127	693	230	25,480.89	8.90	74.74	44.92	Nucleus	41.74	7.83	0.00	50.43
*VaZFP55*	VAG0131613.1	Scaffold_1,769: 9,136..19,344	2535	844	93,149.44	7.82	61.39	53.56	Nucleus	27.01	14.81	0.00	58.18
*VrZFP1*	XP_034681774.1	Chr1: 553,635..556,371	1566	521	57,509.68	8.90	61.94	48.70	Nucleus	22.65	15.16	0.00	62.19
*VrZFP2*	XP_034682326.1	Chr1: 3,258,529..3,261,625	825	274	29,280.15	8.49	44.85	56.43	Nucleus	8.76	9.12	0.00	82.12
*VrZFP3*	XP_034682333.1	Chr1: 3,258,870..3,260,994	822	273	29,209.07	8.49	44.65	56.60	Nucleus	8.79	9.52	0.00	81.68
*VrZFP4*	XP_034695126.1	Chr1: 3,424,130..3,425,166	666	221	23,139.71	8.37	56.24	53.30	Nucleus	21.72	13.57	0.00	64.71
*VrZFP5*	XP_034697189.1	Chr1: 6,102,930..6,106,907	741	246	26,810.93	5.43	63.86	56.18	Nucleus	22.36	12.20	0.00	65.45
*VrZFP6*	XP_034697191.1	Chr1: 6,102,755..6,106,907	741	246	26,810.93	5.43	63.86	56.18	Nucleus	22.36	12.20	0.00	65.45
*VrZFP7*	XP_034674477.1	Chr1: 10,678,040..10,683,798	855	284	32,077.11	9.08	56.69	66.02	Nucleus	18.31	14.44	0.00	67.25
*VrZFP8*	XP_034674418.1	Chr1: 10,678,300..10,679,190	891	296	33,638.13	9.07	59.66	63.17	Nucleus	19.93	16.22	0.00	63.85
*VrZFP9*	XP_034705328.1	Chr2: 11,357,933..11,359,159	747	248	27,154.83	4.96	58.31	64.55	Nucleus	25.40	7.26	0.00	67.34
*VrZFP10*	XP_034703703.1	Chr2: 13,915,433..13,916,772	981	326	35,996.77	5.68	54.51	71.19	Cell membrane	26.38	8.59	0.00	65.03
*VrZFP11*	XP_034682127.1	Chr3: 2,510,558..2,511,978	969	322	35,038.46	7.57	66.09	54.66	Nucleus	18.32	12.73	0.00	68.94
*VrZFP12*	XP_034680378.1	Chr3: 6,228,170..6,233,006	570	189	21,216.78	6.52	56.30	51.58	Nucleus	19.58	6.88	0.00	73.54
*VrZFP13*	XP_034679895.1	Chr3: 6,752,634..6,753,590	708	235	25,608.61	8.53	53.23	71.97	Nucleus	10.64	4.68	0.00	84.68
*VrZFP14*	XP_034682721.1	Chr4: 5,909,744..5,911,153	864	287	31,672.25	5.96	65.33	56.98	Nucleus	23.00	8.36	0.00	68.64
*VrZFP15*	XP_034686446.1	Chr5: 776,318..777,298	708	235	25,349.21	8.87	51.15	73.99	Nucleus	25.11	14.47	0.00	60.43
*VrZFP16*	XP_034686918.1	Chr5: 6,643,815..6,644,992	687	228	25,498.55	8.72	65.39	53.58	Nucleus	31.14	3.95	0.00	64.91
*VrZFP17*	XP_034685738.1	Chr5: 6,656,758..6,657,378	621	206	22,844.47	6.21	64.42	62.93	Nucleus	23.30	0.97	0.00	75.73
*VrZFP18*	XP_034686948.1	Chr5: 6,664,514..6,665,574	537	178	20,294.01	6.53	32.42	62.89	Nucleus	17.42	5.06	0.00	77.53
*VrZFP19*	XP_034688450.1	Chr6: 3,546,885..3,547,993	891	296	31,697.13	6.27	56.05	70.01	Nucleus	7.77	15.54	0.00	76.69
*VrZFP20*	XP_034689063.1	Chr6: 6,443,038..6,444,321	1284	427	46,095.59	9.14	59.34	54.52	Nucleus	9.84	13.35	0.00	76.81
*VrZFP21*	XP_034689064.1	Chr6: 6,451,092..6,452,480	1389	462	50,154.40	9.29	60.39	47.21	Nucleus	22.08	8.66	0.00	69.26
*VrZFP22*	XP_034688666.1	Chr6: 16,774,764..16,776,116	819	272	30,626.48	9.10	60.59	70.40	Nucleus	23.53	13.97	0.00	62.50
*VrZFP23*	XP_034689169.1	Chr6: 16,779,202..16,779,783	582	193	21,182.56	6.10	58.19	37.24	Nucleus	28.50	12.44	0.00	59.07
*VrZFP24*	XP_034688453.1	Chr6: 17,847,675..17,848,394	480	159	17,839.70	9.17	81.51	50.20	Nucleus	28.30	9.43	0.00	62.26
*VrZFP25*	XP_034689071.1	Chr6: 6,514,792..6,516,054	1263	420	45,497.08	9.17	61.29	53.16	Nucleus	11.90	12.14	0.00	75.95
*VrZFP26*	XP_034689322.1	Chr7: 4,267,202..4,268,089	888	295	31,820.29	6.64	47.97	67.37	Nucleus	15.25	14.24	0.00	70.51
*VrZFP27*	XP_034691695.1	Chr7: 4,923,281..4,924,383	582	193	21,442.80	6.16	52.69	45.49	Nucleus	23.32	10.88	0.00	65.80
*VrZFP28*	XP_034689870.1	Chr7: 5,329,636..5,330,682	693	230	24,733.61	7.22	61.61	67.80	Nucleus	8.26	13.04	0.00	78.70
*VrZFP29*	XP_034691020.1	Chr7: 20,623,883..20,626,719	576	191	20,988.50	9.19	58.85	45.75	Nucleus	37.70	8.38	0.00	53.93
*VrZFP30*	XP_034691021.1	Chr7: 20,624,803..20,626,501	576	191	20,988.50	9.19	58.85	45.75	Nucleus	37.70	8.38	0.00	53.93
*VrZFP31*	XP_034691940.1	Chr7: 24,703,955..24,706,387	528	175	19,741.78	4.78	54.11	66.80	Nucleus	20.57	11.43	0.00	68.00
*VrZFP32*	XP_034690590.1	Chr7: 24,711,726..24,712,268	543	180	20,346.73	9.30	56.94	59.24	Nucleus	31.67	9.44	0.00	58.89
*VrZFP33*	XP_034694709.1	Chr8: 15,096,568..15,097,500	561	186	20,496.71	6.14	59.30	48.92	Nucleus	22.04	9.14	0.00	68.82
*VrZFP34*	XP_034693315.1	Chr8: 15,104,445..1,510,5955	912	303	34,090.07	8.70	59.87	69.40	Nucleus	27.39	8.91	0.00	63.70
*VrZFP35*	XP_034694346.1	Chr8: 18,924,297..18,925,593	837	278	30,314.61	6.07	53.78	69.19	Nucleus	10.43	15.47	0.00	74.10
*VrZFP36*	XP_034693265.1	Chr8: 20,767,588..20,768,748	717	238	26,451.18	6.76	53.28	58.92	Nucleus	14.29	13.03	0.00	72.69
*VrZFP37*	XP_034696432.1	Chr9: 11,094,899..11,096,011	804	267	29,396.58	7.10	67.64	56.92	Nucleus	15.73	13.11	0.00	71.16
*VrZFP38*	XP_034700480.1	Chr11: 930,628..933,010	465	154	17,251.04	7.91	62.08	59.60	Nucleus	20.13	16.23	0.00	63.64
*VrZFP39*	XP_034704073.1	Chr13: 7,805,286..7,806,164	879	292	32,002.73	8.61	51.78	64.67	Nucleus	15.75	9.93	0.00	74.32
*VrZFP40*	XP_034703438.1	Chr13: 21,915,467..21,916,664	909	302	33,145.87	6.13	60.86	80.72	Nucleus	20.53	7.62	0.00	71.85
*VrZFP41*	XP_034704234.1	Chr13: 25,793,212..25,794,054	843	280	30,664.17	6.67	65.50	68.04	Nucleus	30. 00	6.79	0.00	63.21
*VrZFP42*	XP_034704827.1	Chr13: 26,761,501..26,762,303	531	176	19,162.04	9.10	71.59	55.76	Nucleus	29.55	10.23	0.00	60.23
*VrZFP43*	XP_034706344.1	Chr14: 154,977..157,422	1713	570	63,789.92	6.39	55.60	57.03	Nucleus	22.98	13.86	0.00	63.16
*VrZFP44*	XP_034708229.1	Chr14: 2,105,931..2,107,900	870	289	32,409.72	8.79	68.55	57.69	Nucleus	30.10	12.46	0.00	57.44
*VrZFP45*	XP_034706389.1	Chr14: 5,129,008..5,129,403	396	131	14,365.07	8.69	60.53	69.37	Nucleus	25.95	6.11	0.00	67.94
*VrZFP46*	XP_034707874.1	Chr14: 10,193,748..10,194,599	633	210	23,213.31	8.83	57.33	60.93	Nucleus	27.62	8.57	0.00	63.81
*VrZFP47*	XP_034708534.1	Chr15: 7,068,109..7,069,605	1080	359	40,180.52	5.36	56.63	69.67	Nucleus	23.40	8.36	0.00	68.25
*VrZFP48*	XP_034708821.1	Chr15: 7,266,510..7,267,953	1404	467	51,080.1	6.76	61.43	47.82	Nucleus	22.06	12.63	0.00	65.31
*VrZFP49*	XP_034711786.1	Chr16: 9,463,294..9,464,251	708	235	26,355.61	8.77	53.62	57.54	Nucleus	30.64	11.91	0.00	57.45
*VrZFP50*	XP_034672939.1	Chr17: 11,002,420..11,003,574	591	196	21,982.62	6.18	60.26	55.26	Nucleus	29.59	7.65	0.00	62.76
*VrZFP51*	XP_034712135.1	Chr17: 12,160,011..12,162,489	1692	563	62,310.44	8.64	57.02	54.64	Nucleus	20.60	14.92	0.00	64.48
*VrZFP52*	XP_034674136.1	Chr18: 7,697,776..7,698,816	702	233	25,322.34	8.91	59.14	74.76	Nucleus	18.03	6.87	0.00	75.11
*VrZFP53*	XP_034674743.1	Chr18: 8,025,926..8,027,417	582	193	21,543.18	8.73	52.12	53.15	Nucleus	24.35	8.81	0.00	66.84
*VrZFP54*	XP_034678325.1	Chr19: 10,956,892..10,959,545	1,836	611	66,157.24	6.28	51.18	43.75	Nucleus	18.99	9.98	0.00	71.03

## Data Availability

Data will be made available on request.

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
