# Peer review of "Genome-Wide Identification and Analysis of the Genes Encoding Q-Type C2H2 Zinc Finger Proteins in Grapevine"

_ijms, 2023, doi:10.3390/ijms242015180_

Round 1

Reviewer 1 Report

Manuscript "Genome-wide identification and analysis of the genes encoding Q-type C2H2 zinc finger proteins in grapevine" is very interesting.

General comments:
Authors performed a genome-wide identification of ZFP genes in the three species of grapevine was based on the current sequence databases. Authors performed phylogenetic and expression analyses.

Detailed comments:
Introduction is good.
Figure 1: What measure was used to calculate the similarity?
Figure 2: What measure was used to calculate the similarity?
Figure 3: The quality of the Figure is very poor. It should be improved.
Figure 4: The quality of the Figure is very poor. It should be improved.
Figure 5: The quality of the Figure is very poor. It should be improved.
Figure 7: What measure was used to calculate the similarity?
Figure S1: What measure was used to calculate the similarity?
Figure S2: What measure was used to calculate the similarity?
Figure S3: What measure was used to calculate the similarity?
Figure 8: The quality of the Figure is very poor. It should be improved. The figure is completely unreadable.
Figure 9: The quality of the Figure is very poor. It should be improved. The figure is completely unreadable.
Figure 10: The quality of the Figure is very poor. It should be improved. The figure is completely unreadable.
Figure 11: The quality of the Figure is very poor. It should be improved. The figure is completely unreadable.

My suggestions:
Latin names should be written in italics. Should be corrected throughout the manuscript.

I have to admit that this is the first time I have read a manuscript on "Genome-wide identification and analysis" that did not estimate and test the effects of parameters on strictly "Genome-wide identification." A paper in this form is not suitable for publication in any scientific journal.
The manuscript should be thoroughly revised. The authors should contact a statistician.

Paper needs major revision.

Author Response

Dear Reviewer,

Thank you very much for your time involved in reviewing the manuscript and your very comments and professional help us to improve the quality of our manuscript. Based on your suggestions and request, we have made corrected modifications on the revised manuscript, the detailed corrections are listed in the attachment.

To facilitate this view, we first retype your comments in italic font with red color and then present our responses to the comments in regular font.   

All response can be found in the attachment.

Reviewer 2 Report

The science of the paper is excellent, the amount of work performed is great... but is a pity that such good work is eclipsed by a very poor and sloppy presentation. Authors should have paid more attention to the formal aspect of the manuscript, as there are crucial mistakes in the presentation.

a) Line numbers should be continuous, and in the present version line number restart in different sections.

b) Lines 34-42 authors have not deleted the template text. Do they really revised the manuscript before submitting? 

c) Line 46: 2+ should be written in superscript.

d) Throughout the manuscript genes are written either in italics or in regular, sometimes even in the same sentence. For instance lines 110-114 and 132-133. Please, use the same criteria (i.e. italics for gene names) in the whole manuscript.

e) A similar problem happens with binnomial names. They should be written in italics, but in many parts of the text (I.e. figure 1 legend) appear like regular font. Please correct. 

f) In page 6, lines138 authors have used italics for the text. Please revise and clean the manuscript. There are also some words underlined out of context (page 6 line 138).

g) Also lettering in figures 8-11 is too small.

So my recommendation is to work on the draft and improve the presentation for a proper evaluation. 

Author Response

Dears reviewer,

Thank you very much for taking the time to review this manuscript to improve the quality of our manuscript. Based on your suggestions and request, we have made corrected modifications on the revised manuscript, the detailed corrections are listed below.

To facilitate this view, we first retype your comments in italic font and then present our responses to the comments in regular font.

All the response can be found in the attachment.

Yours Sincerely

Round 2

Reviewer 1 Report

The authors revised the manuscript according to all comments and suggestions. It can be published in this form.